# PEACE: A Dataset of Pharmaceutical Care for Cancer Pain Analgesia Evaluation and Medication Decision

**Yutao Dou**[1,2]*, **Huimin Yu**[2]*, **Wei Li**[3], **Jingyang Li**[2], **Fei Xia**[1], **Jian Xiao**[2]†

[1] College of Computer Science and Electronic Engineering, Hunan University, Changsha 410082, China.
[2] Department of Pharmacy, Xiangya Hospital, Central South University, Changsha 410008, China.
[3] School of Computer Science, The University of Sydney, Darlington, NSW, 2008, Australia.
ytdou@hnu.edu.cn, 228112395@csu.edu.cn, weiwilson.li@sydney.edu.au,
228112396@csu.edu.cn, xcyphoenix@hnu.edu.cn, admanoas@163.com

## Abstract

Over half of cancer patients experience long-term pain management challenges. Recently, interest has grown in systems for cancer pain treatment effectiveness assessment (TEA) and medication recommendation (MR) to optimize pharmacological care. These systems aim to improve treatment effectiveness by recommending personalized medication plans based on comprehensive patient information. Despite progress, current systems lack multidisciplinary treatment (MDT) team assessments of treatment and the patient's perception of medication, crucial for effective cancer pain management. Moreover, managing cancer pain medication requires multiple adjustments to the treatment plan based on the patient's evolving condition, a detail often missing in existing datasets. To tackle these issues, we designed the PEACE dataset specifically for cancer pain medication research. It includes detailed pharmacological care records for over 38,000 patients, covering demographics, clinical examination, treatment outcomes, medication plans, and patient self-perceptions. Unlike existing datasets, PEACE records not only long-term and multiple follow-ups both inside and outside hospitals but also includes patients' self-assessments of medication effects and the impact on their lives. We conducted a proof-of-concept study with 13 machine learning algorithms on the PEACE dataset for the TEA (classification task) and MR (regression task). These experiments provide valuable insights into the potential of the PEACE dataset for advancing personalized cancer pain management. The dataset is accessible at: [https://github.com/YTYTYD/PEACE].

## 1   Introduction

Cancer pain is a common symptom among cancer patients, with an incidence rate of up to 53%. This greatly affects patients' quality of life and may impede effective cancer treatment. Pharmacotherapy, the mainstay of cancer pain management, often involves long-term medication use. Physicians must continually assess the efficacy of the current analgesic regimen by considering factors such as the patient's physical condition, pain intensity, type of pain, and prior medications. This enables targeted adjustments to the treatment plan to improve therapeutic outcomes.

---

*Equal contribution
†Corresponding author

Submitted to the 38th Conference on Neural Information Processing Systems (NeurIPS 2024) Track on Datasets and Benchmarks. Do not distribute.

Recently, machine learning and deep neural network technologies have significantly advanced automated treatment effect assessment (TEA) and medication recommendation (MR) systems for cancer pain management. These systems use patient data to make accurate assessments and provide medication recommendations. However, most existing systems focus on single treatments and rarely include long-term follow-up. In practice, medication assessment and decision-making often rely on multidisciplinary treatment (MDT) collaboration. Including a pharmacist can significantly enhance cancer pain management efficiency and improve patient pain control and medication adherence. Notably, widely used public datasets like MIMIC [11, 10] and FAERS [28] lack ongoing MDT assessments of patients' medication rationality.

We developed the PEACE (Pharmaceuticals for Easing cAncer pain with CarE) dataset, a comprehensive resource specifically designed for the construction of TEA and MR systems for cancer pain. Compared to other cancer pain related datasets, PEACE offers significant improvements in both the size of patient records and the duration of observations. To our knowledge, it is the first cancer pain medication dataset that provides long-term patient observations and comprehensively contains the information required for MDT decision-making. This dataset includes in-hospital features (patient information, laboratory indicators, physician diagnoses) and out-of-hospital features (patient comments, medication feedback, impact on life). Additionally, it details the MDT's evaluation of the patient's medication use and treatment planning rationale.

Our main contributions are as follows:

1. We release the PEACE dataset[1], the first known resource specifically designed for pharmaceutical care in cancer pain management. This dataset contains over 38,000 patient records, encompassing 103 features related to diverse pathologies, symptoms, and etiologies. It includes multi-visit, long-term observations for 2,600 patients, providing valuable insights into patient care trajectories.

2. PEACE incorporates medical professionals' assessments of the current health state and the rationale behind medication plans, which are not present in existing datasets.

3. We conducted extensive experiments with this dataset, validating the efficacy of 13 machine learning and deep learning approaches in enhancing treatment effect evaluations and medication decision-making.

## 2   Related work

To build reliable TEA and MR systems, it is crucial to gather comprehensive data on both inpatients and outpatients. This includes medication details, treatment outcomes, adverse events and their etiologies, treatment adjustments, and impact on patients' quality of life. However, no public dataset currently meets all these requirements comprehensively. Widely used datasets such as MIMIC-III [11] and MIMIC-IV [10], while detailed in recording medication specifics, lack pharmacist assessments of treatment outcomes. These datasets primarily focus on single hospitalization events rather than the long-term health status of patients, which is particularly disadvantageous for managing chronic conditions like cancer pain. Similarly, the eICU Collaborative Research Database [20] documents essential medication usage information but fails to provide clear explanations of medication effects and lacks long-term patient follow-up. Additionally, these datasets lack patient feedback on their treatment plans. SEER [26] is a representative large-scale cancer registry databases in the United States, compiling extensive retrospective clinical data. It primarily focuses on the treatment processes of cancer patients but does not include assessments of medication plans following hospital discharge. For medication effect assessment, the SIDER [13] database lists adverse reactions for marketed drugs, while the FAERS [28] and TwoSIDES [27] datasets record potential drug interactions. Although these datasets are useful in some aspects, they generally lack detailed records of patients' conditions and necessary clinical features, limiting their practical utility. ISS[19] is a cancer pain assessment dataset that includes videos of 29 patients, along with their self-reported pain scale scores, used to

---

[1]Dataset available at https://github.com/YTYTYD/PEACE

predict the patients' pain levels. A common shortfall of these datasets is their inability to continuously observe and assess patient conditions. They often describe data from a single perspective and fail to integrate the diverse characteristics needed for making MDT decisions. The following section details the PEACE dataset and the steps taken to construct it, aiming to address the deficiencies of existing datasets.

# 3 Dataset Construction

As illustrated in Figure 1, the PEACE dataset construction process begins with clinical data manually collected from hospital, along with follow-up web interactions for patient-reported symptoms. Patient identifiers are anonymized, and dates are shifted to ensure privacy. Feature selection is conducted by experts using the Delphi consensus method [9], a structured communication technique that relies on a panel of experts answering questionnaires in multiple rounds to reach a consensus on key attributes. Data preprocessing involves standardization, imputation, and simplification. Finally, features are categorized, and the processed data is structured into a consistent format, ready for analysis, ensuring both data integrity and privacy protection.

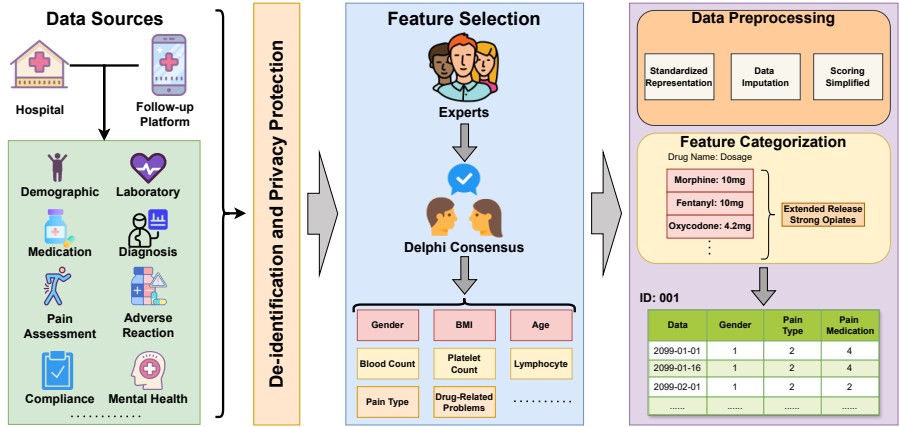

Figure 1: Overview of the data construction process for the PEACE dataset

## 3.1 Data Sources

The data used in this study was collected from two main sources. The first part originated from the Xiangya hospital, encompassing a broad range of patient information, including patient demographics, clinical signs, medication details, physiological parameters, and treatment outcomes. The second part of the data source is our cancer pain online follow-up platform. This platform allows continuous follow-up of cancer pain patients after hospital discharge through patient-initiated reports. It includes patient basic information, pain levels, adverse reactions from medication, dynamic adjustments to medication, treatment of adverse reactions, and other related data. Further details regarding the online follow-up platform can be found in Appendix A.

The inclusion criteria for this research required subjects to have a definitive diagnosis of cancer with associated pain, confirmed via histopathological or cytological methods, with cancer being the primary diagnosis in their medical records. Exclusion criteria included cases with severely incomplete key medical records or significant medical complications. Our work is approved by the Institutional Review Board of the Xiangya Hospital (Ethics Approval ID: 202109422). This work does not interfere with clinical care and treatment procedures. Informed consent is obtained from the patients, and all protected health information is de-identified.

## 3.2 De-identification and Privacy Protection

In the collected data, patient identifiers were removed, and each patient was assigned a unique randomized code ID. Date and time values were shifted 30 to 80 years into the future using a personalized random offset measured in years. Each patient received an independent date transformation, ensuring that the temporal sequence within their data remained consistent. For instance, if the interval between two measurements in the original data was 15 days, the same interval was maintained in the PEACE dataset. However, temporal data for different patients are not comparable. This means that two patients treated in the year 2100 in the dataset are not necessarily treated in the same year in reality. Patients older than 89 years were uniformly labeled as 89 years old to protect their privacy, and patients younger than 18 years were excluded from the dataset. Finally, patient-specific diagnostic reports were reorganized, classified into different categories, and clearly labeled to facilitate data analysis and model training while ensuring privacy protection.

## 3.3 Features Selection

Inspired by [30], this section identifies key features in cancer pain management through the Delphi consensus process, integrating insights from clinical practice and MDT pharmaceutical care. In clinical pharmacy, the Delphi technique is primarily used to develop guidelines or pathways. This is achieved through several rounds of anonymous surveys, repeated consultations, multiple revisions, and generalizations, ultimately leading to the convergence of final opinions [18]. The detailed screening process is outlined in Appendix B.

### 3.3.1 Expert Panel Recruitment

We employed judgmental sampling [2], a targeted recruitment strategy, to identify and invite experts in cancer pain management. Detailed descriptions of the study design and objectives were provided to ensure informed participation. This transparency allowed potential candidates to understand the research goals, methods, and their role in contributing expertise. A multidisciplinary team of experts was assembled to create an effective feature list. This team included clinical pharmacists, anesthetists, oncologists, and nurses. All experts met the following criteria: employment at a tertiary hospital, a minimum of five years of experience in cancer pain management, holding an academic role within a provincial cancer pain association, and willingness to participate in two questionnaire rounds. To ensure balanced representation among professionals, we aimed to maintain equal numbers of doctors and nurses as suggested in [21], with pharmacists serving as additional specialists. We finally recruited 32 experts, including 16 pharmacists, 4 anesthetists, 4 oncologists, and 8 nurses, all based in tertiary hospitals across nine provinces. Their demographics are provided in Appendix B.2.

Experts were required to self-assess their authority (Cr) for each round, based on criteria (Ca) and their familiarity with clinical issues (Cs). The criteria (Ca) encompassed four dimensions: work experience, theoretical analysis, knowledge of domestic and international peers, and insights. Familiarity (Cs) was categorized into five levels: very familiar, familiar, somewhat familiar, unfamiliar, and very unfamiliar, quantified as 1.0, 0.8, 0.6, 0.4, and 0.2, respectively. The questionnaires in both rounds calculated the experts' opinion coordination coefficient (W) and response rate, with a response rate of 75% or higher considered satisfactory. Detailed calculations are provided in Appendix B.1.

### 3.3.2 Delphi Consensus

**The First Round:** In this round, we initiated the Delphi process by inviting experts to participate via email. We informed participants of all study details. The survey began with an introduction and participant demographics section, collecting information like age, gender, education, profession, title, and years of experience. The core of the survey focused on six key themes relevant to cancer pain management pharmaceutical services: patient basic information, comprehensive pain assessment, previous analgesic treatment, evaluation of previous analgesic treatment, cancer pain medication decision, and follow-up. For each theme, experts rated features using a 5-point Likert scale (agreement scale). Additionally, open-ended sections allowed for written feedback.

Following the first round, we calculated average scores and coefficients of variation for each feature. Consensus for an item was defined by meeting the following criteria: 1) average score $\geq 4.0$; 2) coefficient of variation $< 0.15$; and 3) no dissenting opinions. However, if an item received "Agree" or "Strongly Agree" from over 25% of experts but an average score below 3.0, it was carried forward to the second round for further discussion. The first round also encouraged the experts to raise relevant clinical questions. This feedback was collated and shared with all participants as reference material for the second round. Finally, the survey concluded with a self-assessment section where experts rated their own level of expertise and agreement with the overall process. Appendix B.3 provides a more in-depth look at the first round of the Delphi process.

**The Second Round:** This round focused exclusively on features that lacked clear consensus in the first round [1]. Experts received their individual scores alongside the overall distribution and percentages of scores from their peers [24, 25]. This facilitated informed reflection and potential adjustments to their initial ratings. We also considered expert suggestions for modifying existing questions or introducing new ones from the first round. These were incorporated into personalized questionnaires for the second round. Stringent inclusion criteria remained for the second round. Features required an average rating of at least 4.0 (strongly agree), and a coefficient of variation less than 0.15 (low variability) to be considered for the final list. Please see Appendix B.4 for a detailed breakdown of the second-round process.

## 3.4 Data Preprocessing

**Data Standardization:** The raw medication data presented significant challenges for direct modeling due to noise, complex attribute relationships, and high dimensionality. Common issues included disorganization, duplicate records, and missing information, which complicate model training. To mitigate these challenges, we implemented a comprehensive data preprocessing pipeline. For example, we standardized synonym variations within pain intensity labels. Terms like "burning pain," "scalding pain," and "burn-like pain" were standardized to "burning-type pain" to ensure consistent representation. Redundancies were addressed by merging useful fields from duplicate records to enhance data quality. For data inconsistencies and anomalies potentially arising from human errors, we employed a two-pronged approach. When sample sizes permitted, we opted for data correction through expert consultation to preserve valuable information. In cases where data accuracy could not be confirmed, or sample sizes were inadequate, data points were removed to prevent model bias and improve training robustness.

**Feature Categorization:** The original data included numerous multiple-choice features, such as various analgesics with similar effects but different brands or specifications. Patients might also take several similar drugs simultaneously due to complementary effects. Given the large number of possible combinations, directly including these features in the model may lead to suboptimal performance. To mitigate this, we categorized these features to structure them for better usability in machine learning tasks. For instance, combinations of dozens of drugs in the raw data were grouped into seven categories based on their actions and specifications: "Extended Release Strong Opiates (ERSO)," "Immediate Release Strong Opiates (IRSO)," "Extended Release Weak Opiates (ERWO)," "Immediate Release Weak Opiates (IRWO)," "Nonsteroidal Anti-Inflammatory Drugs (NSAID)," "Anticonvulsants/Antidepressants (A/A)," and "Others," with numerical representation of the quantity of medication used per category. Similarly, we classified patients' pain types into four categories by integrating specific pain locations, pain intensity, and the nature of the pain, providing the model with a comprehensive representation of pain characteristics. Additionally, we addressed the high dimensionality of the pain intensity score. The original specific number of times or persistent pain was simplified into a more practical multiclassification (0: none, 1: $<3$ times, 2: $\geq 3$ times, and 3: persistent pain) to improve model efficiency without compromising essential information.

## 3.5 Dataset Features

Our data construction process resulted in a comprehensive dataset encompassing 103 features, broadly categorized into six groups. The Patient Baseline Information group (50 features) captures demographic and clinical characteristics of the patients, potentially including age, gender, co-morbidities, and disease stage. The Comprehensive Pain Assessment group (15 features) details the extent and characteristics of the patients' pain experience, potentially including pain intensity scores, pain quality descriptors (e.g., visceral pain, somatic pain), and functional limitations. The Previous Analgesic Treatment group (23 features) details the medications and interventions previously used to manage the patients' pain, potentially including medication names, dosages, durations, and routes of administration. The Evaluation of Previous Analgesic Treatment group (5 features) captures the effectiveness and tolerability of prior pain management strategies, potentially including patient-reported outcomes or physician assessments. The Cancer Pain Medication Decision group (9 features) details the rationale behind the selection of specific pain medications for the study participants, potentially including factors like pain type, treatment history, and co-morbidities. The Follow-Up group (1 feature) captures information on patient outcomes after the intervention of interest, potentially including pain response or adverse events. A detailed description of each feature is provided in Appendix B.5.

## 3.6 Dataset Descriptive Analysis

**Feature distribution:** Table 1 categorizes the 103 features in the PEACE dataset, with numeric features comprising the majority at 75%.

| Patient Basic Information | | | | Comprehensive Pain Assessment | | | | Previous Analgesic Treatment | | | |
|---|---|---|---|---|---|---|---|---|---|---|---|
| Total | Binary | Multiclass | Numerical | Total | Binary | Multiclass | Numerical | Total | Binary | Multiclass | Numerical |
| 50 | 6 | 2 | 42 | 15 | 0 | 4 | 11 | 23 | 5 | 0 | 18 |
| **Evaluation of Previous Analgesic Treatment** | | | | **Cancer Pain Medication Decision** | | | | **Follow-up** | | | |
| Total | Binary | Multiclass | Numerical | Total | Binary | Multiclass | Numerical | Total | Binary | Multiclass | Numerical |
| 5 | 0 | 5 | 0 | 9 | 2 | 0 | 7 | 1 | 0 | 1 | 0 |

Table 1: Summary of dataset features distribution.

**Demographics:** The socio-demographic statistics of our patients are presented in Figure 2 (a), showing that the 45-74 age group has the highest cancer incidence. Figure 2 (b) illustrates the gender distribution, which is nearly balanced with a male-to-female ratio of 51.4:48.6. See Appendix C for more detailed demographics.

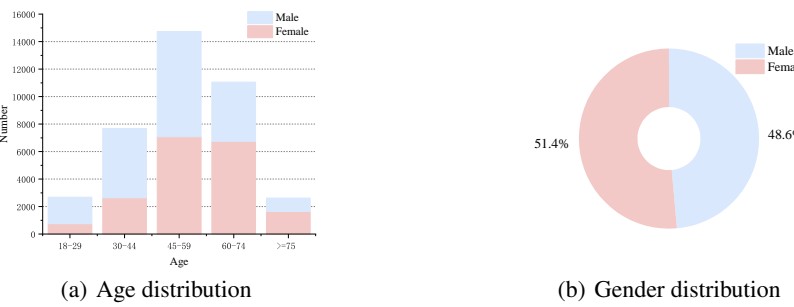

(a) Age distribution  (b) Gender distribution

Figure 2: Patient demographics: Age and gender distribution

**Visit Statistics:** Table 2 summarizes patient visit statistics. Notably, 7% of patients have multiple visits recorded, with a maximum of 33 visits.

**Patient Sample:** We present a sample patient with selected features from the PEACE dataset in Table 3. The table illustrates how medical staff adjust the patient's medication based on the effectiveness of each treatment and the drug reactions experienced during the medication process. This approach

Table 2: Statistics on the patient records

|  | Number of patients | Avg | Std dev | Min | 1st quartile | Median | 3rd quartile | Max |
|---|---|---|---|---|---|---|---|---|
| All patients | 38,766 | 1.09 | 0.58 | 1 | 1 | 1 | 1 | 33 |
| Patients with records $\geq$ 2 | 2,601 | 2.48 | 1.74 | 2 | 2 | 2 | 2 | 33 |
| Patients with records $\geq$ 3 | 514 | 4.44 | 3.27 | 3 | 3 | 3 | 4 | 33 |
| Patients with records $\geq$ 5 | 116 | 8.69 | 4.86 | 5 | 6 | 7 | 9.25 | 33 |
| Patients with records $\geq$ 10 | 29 | 14.82 | 6.25 | 10 | 11 | 13 | 16 | 33 |

aims to mitigate adverse reactions and achieve better outcomes. The complete data for this patient and additional patient samples are provided in Appendix F.

Table 3: A sample patient from the PEACE dataset (Pain Relief and Post-medication Pain Score: 1. Complete Relief, 2. Partial Relief, 3. Mild Relief, 4. Ineffective)

**ID: SJ-289031**

| Patient Basic Information | | | | | | | |
|---|---|---|---|---|---|---|---|
| Gender | Age | Length of Hospital Stay | Discharge Diagnosis | Smoking History | Treatment Method | White Blood Cell Count | Total 50 Features |
| 1 | 59 | 1 | 112 | 0 | 2 | 7.5 | |
| 1 | 59 | 3 | 112 | 0 | 2 | 4.2 | |
| 1 | 59 | 10 | 112 | 0 | 2 | 5.6 | |
| 1 | 59 | 17 | 112 | 0 | 2 | 4.7 | |

| Comprehensive Pain Assessment | | | | | | | |
|---|---|---|---|---|---|---|---|
| Pain Type | Worst Pain | Current Pain | Daily Life | Pain Frequency | Breakthrough Pain Type | Breakthrough Pain Frequency | Total 15 Features |
| 2 | 6 | 6 | 7 | 2 | 2 | 1 | |
| 2 | 4 | 3 | 3 | 2 | 2 | 1 | |
| 2 | 2 | 1 | 1 | 1 | 2 | 2 | |
| 2 | 0 | 0 | 1 | 0 | 0 | 0 | |

| Previous Analgesic Treatment | | | | | | | |
|---|---|---|---|---|---|---|---|
| Days of Medication Use | MMAS-8 Total Score | Prev_ERSO | Prev_IRSO | Prev_NSAID | Duration of Analgesic Control | Nausea or Vomiting | Total 23 Features |
| 3 | 5.75 | 1 | 0 | 0 | 6 | 1 | |
| 5 | 8 | 1 | 0 | 1 | 8 | 1 | |
| 12 | 8 | 1 | 0 | 1 | 8 | 0 | |
| 19 | 8 | 1 | 1 | 1 | 12 | 0 | |

| Cancer Pain Medication Decision | | | | | | | |
|---|---|---|---|---|---|---|---|
| ERSO_Recom | IRSO_Recom | LWO_Recom | IRWO_Recom | NAD_Recom | A/A_Recom | Constipation Management Medication | Total 9 Features |
| 1 | 0 | 0 | 0 | 1 | 0 | 2 | |
| 1 | 0 | 0 | 0 | 1 | 0 | 2 | |
| 1 | 0 | 0 | 0 | 1 | 0 | 2 | |
| 1 | 0 | 0 | 0 | 1 | 0 | 2 | |

| Evaluation of Previous | | | | | | Follow-up | |
|---|---|---|---|---|---|---|---|
| Drug-Related Problems | Causes | Interventions | Acceptance of Interventions | Status of DRPs | | Pain Relief and Post-medication Pain Score | |
| 2 | 1 | 15 | 1 | 3 | | 3 | |
| 2 | 9 | 11 | 1 | 3 | | 2 | |
| 2 | 9 | 10 | 1 | 3 | | 1 | |
| 0 | 0 | 0 | 0 | 0 | | 1 | |

## 3.7 Dataset Usage

The PEACE dataset is designated for research purposes exclusively. The dataset access process involves three steps: 1) Completing relevant training (such as the CITI or GCP training), 2) Signing and adhering to a data use agreement, and 3) Obtaining approval from Xiangya Hospital. The agreement outlines responsible data handling practices and emphasizes the importance of following established collaborative research ethics. Models trained on this dataset should undergo rigorous evaluation before real-world deployment. This evaluation should assess the model's performance, generalizability, and representativeness for the target real-world application. A detailed description of the PEACE dataset usage is provided in Appendix E.

# 4 Experiment

## 4.1 Experimental Setup

### 4.1.1 Tasks

To establish the TEA/MR system, this study quantitatively assess patient treatment outcomes and guide future treatment strategies. Our PEACE dataset supports two types of prediction tasks: (1) TEA, which is a multi-label classification (levels 1-4) using patient characteristics with time series data to quantify levels of treatment efficacy; and (2) MR, which involves regression analyses utilizing time series data to predict the quantity of various analgesics required by patients following adjustments in their treatment plans based on their medication history.

### 4.1.2 Baselines

We present the results for 13 algorithms, which cover machine learning and deep learning algorithms, on the PEACE dataset for both tasks. These algorithms include 5 basic machine learning and neural network models: Decision Trees [22], Logistic Regression [5], Random Forests [14], SVM [4] and MLP [23]; 3 popular gradient boosting decision tree methods: LightGBM [12], XGBoost [3], and AdaBoost [6]; 3 advanced neural network models designed for time-series data: iTransformer [15], TransTab [29], and Mamba [8]; and 2 neural network models specifically tailored for electronic health records (EHR): Stagenet[7] and Adacare[16]. Details of the baselines are provided in Appendix D.1.

### 4.1.3 Experiment Environment

In our experiments, 80% of the dataset was used for model building with 5-fold cross-validation, while the remaining 20% served as an independent test set. For detailed information on data partitioning, please refer to Appendix D.2. A random state of 42 was used in all our experiments. The models were trained on a computing platform platform equipped with an Intel i7-13700KF CPU, 128GB of memory, and an NVIDIA RTX4090 24GB GPU.

### 4.1.4 Evaluation Metrics

In our experiments, we used the following metrics to evaluate the performance. For TEA (classification tasks), we used the metrics of accuracy (ACC), area under the receiver operating characteristic curve (AUROC), F1 score, recall, and precision. For MR (regression tasks), we used mean squared error (MSE) and mean absolute error (MAE). The details of the metrics are given in Appendix D.3.

## 4.2 Results

For the TEA task, as shown in Table 4, the GBDT algorithm LightGBM achieved the highest ACC and Recall. This success is due to its ability to handle large-scale, high-dimensional data, robust feature selection, and effective regularization to prevent overfitting. XGBoost also performed well, closely following LightGBM. Basic models like Decision Trees and Logistic Regression, although simple and efficient, struggled with complex data patterns and multidimensional features. General neural network models required more precise tuning and did not perform as well on the tabular format of the PEACE dataset. In contrast, EHR-specific models were better at identifying task-relevant features, leading to improved performance. Detailed results for the K-fold and independent test set experiments for the TEA task are given in Table 14 of Appendix D.4.

For the MR task, as shown in Table 5, tree-based models, including decision trees, random forests, and GBDT, demonstrated good performance and stability, achieving the top results in most metrics. Advanced neural network models like iTransformer, while excelling in specific categories, were prone to overfitting and lacked the robustness of tree-based models. Similar to their performance in the TEA task, neural network models optimized for the EHR scenario show potential for significant improvement. Detailed results on the K-fold and independent test set experiments for the MR task are given in Table 15 of Appendix D.4.

In conclusion, tree-based models, particularly GBDT, performed exceptionally well on the PEACE dataset, which is a typical structured tabular dataset. These models excel in handling irregularities such as skewed and heavy-tailed feature distributions and have strong feature selection capabilities and built-in regularization techniques that prevent overfitting. In contrast, neural networks require extensive tuning, complex architecture designs, and additional regularization measures, making tree-based models more stable and reliable in most cases, as supported by the findings in [17]. Nonetheless, neural networks can achieve performance gains in specialized applications by optimizing their structure, such as incorporating modules for EHR prediction tasks to highlight correlations between critical medical features.

Table 4: TEA Model Performance (The values represent the mean results of 5-fold cross-validation runs and their mean errors.)

| Model | Accuracy | F1 Score | Recall | Precision | AUROC |
|---|---|---|---|---|---|
| Decision Tree | 0.7189 ± 0.0030 | 0.6622 ± 0.0035 | 0.6645 ± 0.0037 | 0.6601 ± 0.0042 | 0.7778 ± 0.0025 |
| Logistic Regression | 0.6780 ± 0.0022 | 0.6040 ± 0.0015 | 0.5738 ± 0.0018 | 0.6776 ± 0.0052 | 0.7202 ± 0.0011 |
| Random Forest | 0.7846 ± 0.0014 | 0.7374 ± 0.0016 | 0.7001 ± 0.0020 | 0.8108 ± 0.0010 | 0.8071 ± 0.0010 |
| SVM | 0.6648 ± 0.0025 | 0.5683 ± 0.0024 | 0.5465 ± 0.0025 | 0.6584 ± 0.0058 | 0.7026 ± 0.0011 |
| MLP | 0.7374 ± 0.0033 | 0.6801 ± 0.0023 | 0.6730 ± 0.0023 | 0.6896 ± 0.0048 | 0.7852 ± 0.0015 |
| XGBoost | 0.7947 ± 0.0023 | 0.7504 ± 0.0024 | 0.7186 ± 0.0033 | 0.8023 ± 0.0027 | 0.8185 ± 0.0015 |
| LightGBM | **0.8023 ± 0.0024** | 0.7616 ± 0.0023 | **0.7297 ± 0.0028** | 0.8155 ± 0.0031 | 0.8259 ± 0.0015 |
| AdaBoost | 0.6647 ± 0.0011 | 0.5596 ± 0.0031 | 0.5513 ± 0.0038 | 0.6321 ± 0.0073 | 0.7043 ± 0.0027 |
| Transtab | 0.5835 ± 0.0034 | 0.3170 ± 0.0029 | 0.3449 ± 0.0026 | 0.3479 ± 0.0357 | 0.6595 ± 0.0027 |
| iTransformer | 0.6606 ± 0.0396 | 0.6456 ± 0.0437 | 0.6608 ± 0.0397 | 0.6506 ± 0.0413 | 0.7036 ± 0.0203 |
| Mamba | 0.7272 ± 0.0281 | 0.7212 ± 0.0322 | 0.7272 ± 0.0282 | 0.7352 ± 0.0259 | 0.7686 ± 0.0200 |
| StageNet | 0.7291 ± 0.0177 | **0.7893 ± 0.0166** | 0.7208 ± 0.0116 | **0.8725 ± 0.0235** | 0.7443 ± 0.0033 |
| AdaCare | 0.7507 ± 0.0111 | 0.7087 ± 0.0107 | 0.6612 ± 0.0326 | 0.7646 ± 0.0061 | **0.8515 ± 0.0047** |

Table 5: MR Model Performance (The values represent the mean results of 5-fold cross-validation runs and their mean errors.)

| Model | ERSO MSE | ERSO MAE | IRSO MSE | IRSO MAE | ERWO MSE | ERWO MAE | IRWO MSE | IRWO MAE | NSAID MSE | NSAID MAE | A/A MSE | A/A MAE | Others MSE | Others MAE |
|---|---|---|---|---|---|---|---|---|---|---|---|---|---|---|
| Decision Tree | 0.0373 ±0.0012 | 0.0360 ±0.0011 | 0.0289 ±0.0014 | 0.0289 ±0.0013 | 0.0101 ±0.0006 | 0.0101 ±0.0006 | 0.0156 ±0.0004 | 0.0156 ±0.0004 | 0.1404 ±0.0015 | 0.1397 ±0.0015 | 0.0328 ±0.0007 | 0.0326 ±0.0007 | 0.0026 ±0.0004 | 0.0023 ±0.0002 |
| Logistic Regression | 0.1278 ±0.0012 | 0.1271 ±0.0013 | 0.1142 ±0.0021 | 0.1142 ±0.0021 | 0.0205 ±0.0007 | 0.0205 ±0.0007 | 0.0426 ±0.0011 | 0.0426 ±0.0011 | 0.0958 ±0.0009 | 0.0954 ±0.0007 | 0.0224 ±0.0008 | **0.0224 ±0.0008** | 0.0020 ±0.0003 | 0.0017 ±0.0003 |
| Random Forest | 0.0189 ±0.0005 | 0.0380 ±0.0008 | 0.0155 ±0.0007 | 0.0311 ±0.0010 | **0.0056 ±0.0002** | 0.0114 ±0.0003 | 0.0082 ±0.0005 | 0.0164 ±0.0006 | **0.0706 ±0.0007** | 0.1405 ±0.0010 | 0.0161 ±0.0006 | 0.0326 ±0.0004 | 0.0013 ±0.0002 | 0.0022 ±0.0002 |
| SVM | 0.1704 ±0.0068 | 0.2722 ±0.0023 | 0.1663 ±0.0039 | 0.1613 ±0.0028 | 0.0310 ±0.0007 | 0.0310 ±0.0007 | 0.0955 ±0.0011 | 0.0956 ±0.0011 | 0.1307 ±0.0016 | **0.1195 ±0.0010** | 0.0432 ±0.0005 | 0.0452 ±0.0013 | 0.0014 ±0.0002 | 0.0013 ±0.0001 |
| MLP | 0.0371 ±0.0016 | 0.1149 ±0.0018 | 0.0316 ±0.0006 | 0.1060 ±0.0014 | 0.0120 ±0.0013 | 0.0561 ±0.0040 | 0.0157 ±0.0007 | 0.0664 ±0.0016 | 0.0869 ±0.0015 | 0.1939 ±0.0020 | 0.0218 ±0.0010 | 0.0760 ±0.0020 | 0.0020 ±0.0001 | 0.0207 ±0.0006 |
| XGBoost | 0.0210 ±0.0006 | 0.0526 ±0.0012 | 0.0165 ±0.0007 | 0.0420 ±0.0008 | 0.0068 ±0.0003 | 0.0164 ±0.0004 | 0.0091 ±0.0005 | 0.0278 ±0.0004 | 0.0725 ±0.0005 | 0.1501 ±0.0007 | 0.0174 ±0.0007 | 0.0364 ±0.0005 | 0.0016 ±0.0002 | 0.0027 ±0.0002 |
| LightGBM | **0.0189 ±0.0004** | 0.0447 ±0.0005 | **0.0154 ±0.0007** | 0.0367 ±0.0008 | **0.0056 ±0.0002** | 0.0136 ±0.0004 | 0.0081 ±0.0005 | 0.0193 ±0.0004 | 0.0674 ±0.0005 | 0.1371 ±0.0006 | **0.0154 ±0.0005** | 0.0314 ±0.0004 | 0.0013 ±0.0002 | 0.0035 ±0.0001 |
| AdaBoost | 0.1913 ±0.0099 | 0.4247 ±0.0139 | 0.0763 ±0.0141 | 0.2178 ±0.0343 | 0.0165 ±0.0019 | 0.0432 ±0.0065 | 0.0497 ±0.0012 | 0.1155 ±0.0020 | 0.2341 ±0.0031 | 0.4831 ±0.0033 | 0.0773 ±0.0363 | 0.1902 ±0.0721 | 0.0427 ±0.0149 | 0.1289 ±0.0457 |
| Transtab | 0.2828 ±0.0012 | 0.2818 ±0.0014 | 0.2330 ±0.0021 | 0.2329 ±0.0022 | 0.0298 ±0.0006 | 0.0302 ±0.0004 | 0.0798 ±0.0004 | 0.0797 ±0.0004 | 0.2940 ±0.0009 | 0.2928 ±0.0004 | 0.0434 ±0.0014 | 0.0430 ±0.0013 | 0.0012 ±0.0002 | 0.0011 ±0.0001 |
| iTransformer | 0.0442 ±0.0134 | 0.0808 ±0.0152 | 0.0537 ±0.0151 | 0.1091 ±0.0203 | 0.0184 ±0.0110 | 0.0400 ±0.0172 | **0.0078 ±0.0016** | 0.0384 ±0.0080 | 0.0867 ±0.0074 | 0.1498 ±0.0182 | 0.1715 ±0.1496 | 0.0539 ±0.0097 | 0.0020 ±0.0007 | 0.0053 ±0.0010 |
| Mamba | 0.0313 ±0.0118 | 0.0526 ±0.0140 | 0.0214 ±0.0088 | 0.0373 ±0.0133 | 0.0243 ±0.0132 | 0.0247 ±0.0148 | 0.0134 ±0.0057 | 0.0254 ±0.0100 | 0.0770 ±0.0231 | 0.1132 ±0.0271 | 0.0210 ±0.0022 | 0.0426 ±0.0037 | 0.0452 ±0.0439 | 0.0055 ±0.0015 |
| StageNet | 0.0297 ±0.0042 | 0.0756 ±0.0021 | 0.1798 ±0.0027 | 0.3585 ±0.0027 | 0.2024 ±0.0007 | 0.4048 ±0.0008 | 0.0823 ±0.0013 | 0.1744 ±0.0012 | 0.2098 ±0.0017 | 0.4149 ±0.0020 | 0.0399 ±0.0015 | 0.0816 ±0.0014 | **0.0005 ±0.0002** | **0.0005 ±0.0002** |
| AdaCare | 0.0246 ±0.0031 | **0.0281 ±0.0027** | 0.0165 ±0.0009 | **0.0194 ±0.0007** | 0.0081 ±0.0013 | **0.0090 ±0.0016** | 0.0116 ±0.0013 | **0.0139 ±0.0013** | 0.0993 ±0.0029 | 0.1196 ±0.0034 | 0.0257 ±0.0021 | 0.0279 ±0.0022 | **0.0004 ±0.0002** | 0.0005 ±0.0003 |

## 5 Limitation

We acknowledge the following limitations: First, the expert consensus-derived features were obtained from experts across only nine provinces in China, introducing potential subjectivity to our findings. Second, the dataset comprises 38,000 patients, which may limit the generalizability and performance of the models. Additional samples would be necessary to validate and enhance our findings. Lastly, the models tested in this study have not yet been applied in a clinical setting, leaving their practical efficacy uncertain.

## 6 Conclusion and Future Work

In this work, we introduce PEACE, a comprehensive dataset for cancer pain medication therapy, which comprises over 38,000 patients experiencing cancer-related pain, including more than 2,600 patients with multiple long-term follow-up records. The dataset integrates features from hospital and online follow-up platform through an expert Delphi consensus process. These features encompass demographics, laboratory tests, pain assessments, medication treatments, and variables related to outcome evaluation and medication recommendations. Using this dataset, we evaluated the performance of 13 models on the classification and regression tasks. Our results indicate that existing models are unable to fully harness the dataset's potential. Constructed from a multidisciplinary therapeutic research perspective, PEACE thoroughly incorporates the specifics of the medical field, making it a valuable resource for researchers seeking to extract meaningful medical information. This dataset could be utilized in many studies concerning cancer pain.

In the next phase of our work, we will continue to incorporate more patient information into our dataset to enhance its generalizability and representativeness. We also plan to expand our selected features, particularly with more detailed laboratory indicators such as blood drug concentrations, based on further expert advice. Additionally, we intend to explore potential correlations between human genes, drug molecules, and cancer pain from the perspectives of biogenetics, bioinformatics, and medicinal chemistry to enhance medication safety for patients and reduce adverse effects. This approach will help enhance medication safety and reduce adverse effects. Finally, we will validate our models in clinical settings to assess their practical efficacy and reliability.

## Acknowledgment

This work was supported by the Natural Science Foundation of Hunan Province (No. 2022JJ80114); The Fundamental Research Funds for the Central South University (2024ZZTS0287); Graduate Research Innovation Project of Hunan Province (CX20240417).

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

## A  Online Follow-up Platform

As illustrated in Figure 3, the cancer pain online follow-up platform allows patients to proactively report their condition after hospital discharge. Given that our system operates in a non-English environment, we have translated its pages into English to ensure readability and comprehension.

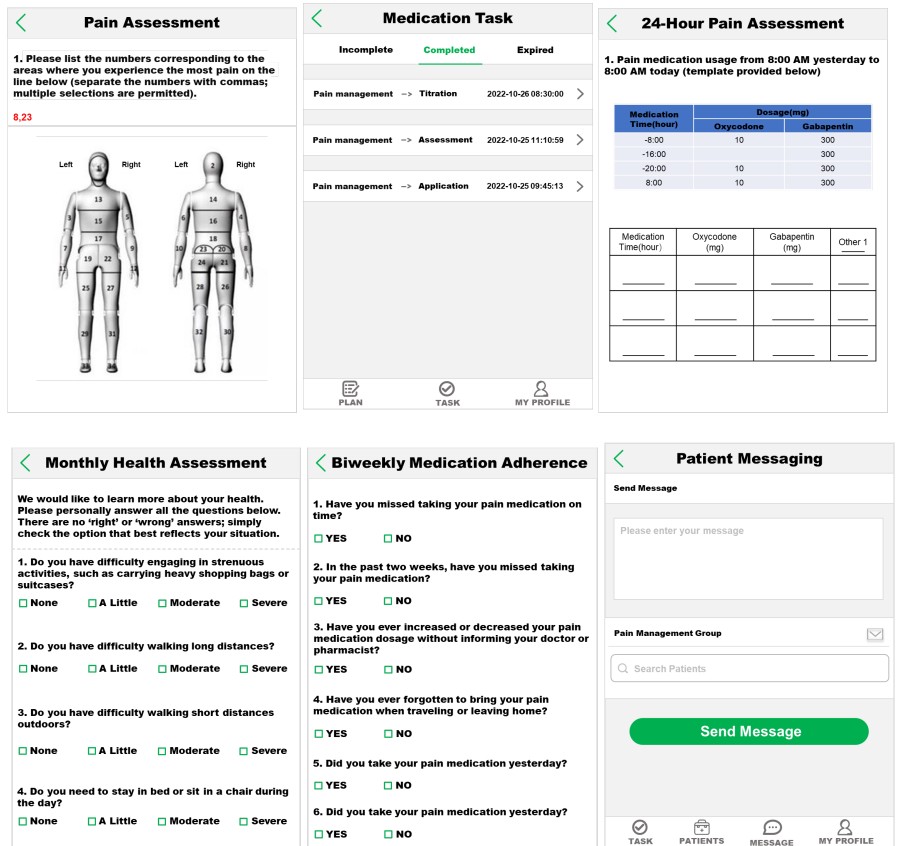

Figure 3: Functions of the cancer pain online follow-up platform (English translation version)

## B  Delphi Process Design

### B.1  Process Design

In each round of the Delphi survey, experts were asked to rate each item using a five-point Likert scale (ranging from strongly agree, agree, neutral, disagree, to strongly disagree). Consensus was defined as: 1) an average score of $\geq 4.0$; 2) a coefficient of variation <0.15; and 3) no dissenting opinions. Additionally, experts were required to self-assess their authority (Cr) for each round, determined by the judgement criteria (Ca) and their familiarity with the clinical issues (Cs). The Ca encompassed four dimensions: work experience, theoretical analysis, understanding from domestic and international peers, and insights. The Cs included five levels: very familiar, familiar, somewhat familiar, unfamiliar, and very unfamiliar, quantified as 1.0, 0.8, 0.6, 0.4, and 0.2, respectively. Both rounds of questionnaires will calculate the experts' coordination coefficient (W) and response rate, with a response rate of 75% or above considered satisfactory. The questionnaires were distributed to experts via email. To ensure a high response rate, each Delphi round was open for two weeks, with email reminders sent at the start and end of each round.

The expert response rate was calculated as follows:

$$\text{Expert Coefficient} = \left( \frac{\text{Number of returned questionnaires}}{\text{Number of distributed questionnaires}} \right) \times 100\% \qquad (1)$$

The coordination ratio $Cr$ was calculated using:

$$Cr = \frac{Ca + Cs}{2} \qquad (2)$$

The experts' opinion coordination coefficient (W) was represented by Kendall's $W$, with differences assessed using the Chi-square ($\chi^2$) test. A $p$-value of less than 0.05 was considered statistically significant.

## B.2 Expert Invitation

A total of 32 experts from nine provinces in China were invited to participate in this study, including 16 pharmacists, 4 anesthetists, 4 oncologists, and 8 nurses. All experts are employed at top-tier hospitals in China. Detailed demographic information of the experts is provided in Table 6.

Table 6: Baseline characteristics of the experts

| Characteristic | N | % |
|---|---|---|
| **Gender** | | |
| Male | 6 | 18.6 |
| Female | 26 | 81.4 |
| **Age** | | |
| 30-39 | 10 | 31.3 |
| 40-49 | 16 | 50.0 |
| $\geq$50 | 6 | 18.7 |
| **Profession** | | |
| Pharmacist | 16 | 50.0 |
| Anaesthetists | 4 | 12.5 |
| Oncologists | 4 | 12.5 |
| Nurse | 8 | 25.0 |
| **Professional title** | | |
| Director | 9 | 28.1 |
| Associate director | 23 | 71.9 |
| **Highest level of education** | | |
| Bachelor degree | 9 | 28.1 |
| Master degree | 12 | 37.5 |
| Doctoral degree | 11 | 34.4 |
| **Experience in cancer pain management (years)** | | |
| 5-9 | 13 | 40.6 |
| 10-19 | 14 | 43.8 |
| 20-29 | 3 | 9.4 |
| $\geq$30 | 2 | 6.2 |

## B.3 First Round Delphi

In the first round of the Delphi survey, experts were invited to rate 21 items across 6 themes, as shown in Table 7(Clinical features of the first round). All items were rated as "Agree" or "Strongly Agree," with an average score of $\geq$4.0. In this round, consensus was reached for 17 items (80.9%) submitted to the expert panel. Specifically, 5 items from Theme A, 5 items from Theme B, 4 items from Theme C, and all items from Themes D, E, and F achieved consensus. Items A3 (Smoking history, alcohol consumption history, allergic history), B6-1 (Worsening factors, including activities,

weather, and mental factors), B6-2 (Alleviating factors, including rest, suitable environment, and taking analgesics), and C2 (Duration of analgesics use) did not meet the inclusion criteria for the coefficient of variation and will thus proceed to the second round.

Additionally, three supplementary items submitted by the experts will be included in the second round: O1 (Monitoring and management of analgesic-related adverse reactions), O2 (Patient lifestyle), and O3 (Drug accessibility).

## B.4 Second Round Delphi

Based on the results of the first round of evaluations, the new questionnaire includes 7 items. In this round, consensus was achieved for 3 items (42.8%) submitted to the expert panel. Items A3, C2, and the newly introduced item O1 were included, while the other items were excluded. The results of the second round are shown in Table 7(Clinical features of the second round).

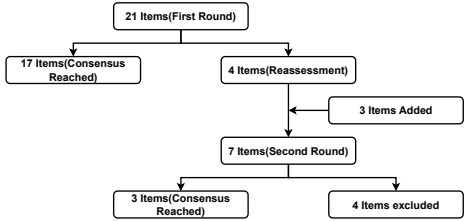

Figure 4: Overview of the Delphi rounds

Table 7: Clinical features of the first and second round

**Clinical features of the first round**

| NO | Clinical Features | Average score | Coefficient of variation (%) | Reach a consensus |
|---|---|---|---|---|
| **Theme A: Patient Basic Information** | | | | |
| A1 | Patient general information and clinical diagnosis | 4.50 | 13.83 | YES |
| A2 | Laboratory examination (including complete blood count, liver function, kidney function) | 4.25 | 13.36 | YES |
| A3 | Smoking history, alcohol consumption history, allergic history | 4.34 | 19.05 | NO |
| A4 | Tumor-related treatment | 4.65 | 12.95 | YES |
| A5 | Performance status | 4.09 | 11.37 | YES |
| A6 | Analgesic risk assessment | 4.71 | 14.48 | YES |
| **Theme B: Comprehensive pain assessment** | | | | |
| B1 | Pain type | 4.62 | 11.97 | YES |
| B2 | Pain intensity, assessed by quantitative tools | 4.78 | 10.27 | YES |
| B3 | Pain frequency | 4.56 | 11.05 | YES |
| B4 | Breakthrough pain assessment | 4.59 | 14.48 | YES |
| B5 | Impact of pain on daily life | 4.43 | 13.95 | YES |
| B6 | Pain worsening or alleviating factors | | | |
| B6-1 | Worsening factors, including activities, weather, and mental factors | 4.37 | 17.17 | NO |
| B6-2 | Alleviating factors, including rest, suitable environment, and taking analgesics | 4.15 | 20.37 | NO |
| **Theme C: Previous analgesic treatment** | | | | |
| C1 | Types of analgesics | 4.81 | 8.24 | YES |
| C2 | Duration of analgesics use | 4.31 | 19.03 | NO |
| C3 | Opioid tolerance | 4.59 | 13.38 | YES |
| C4 | Medication adherence | 4.46 | 13.90 | YES |
| C5 | Analgesic efficacy assessment | 4.28 | 14.81 | YES |
| **Theme D: Evaluation of previous analgesic treatment** | | | | |
| D1 | Including analysis of existing/potential Drug-Related Problems (DRPs), their causes, interventions, and outcomes in previous medication | 4.62 | 14.27 | YES |
| **Theme E: Cancer Pain Medication Decision** | | | | |
| E1 | Cancer pain medication decision based on comprehensive pain assessment | 4.84 | 7.62 | YES |
| **Theme F: Follow-up** | | | | |
| F1 | Pain relief assessment | 4.71 | 11.08 | YES |

**Clinical features of the second round**

| NO | Clinical Features | Average score | Coefficient of variation (%) | Reach a consensus |
|---|---|---|---|---|
| **Theme A3** | | | | |
| | Smoking history, alcohol consumption history, allergic history | 4.15 | 12.39 | YES |
| **Theme B6-1** | | | | |
| | Worsening factors, including activities, weather, and mental factors | 4.21 | 16.74 | NO |
| **Theme B6-2** | | | | |
| | Alleviating factors, including rest, suitable environment, and taking analgesics | 4.25 | 17.93 | NO |
| **Theme C2** | | | | |
| | Duration of analgesics use | 4.68 | 13.75 | YES |
| **Other O1** | | | | |
| | Monitoring and management of analgesic-related adverse reactions | 4.84 | 9.25 | YES |
| **Other O2** | | | | |
| | Drug accessibility | 4.12 | 22.83 | NO |
| **Other O3** | | | | |
| | Lifestyle of patients | 4.25 | 19.82 | NO |

The response rate for both rounds was 100% (32/32). In both rounds of the Delphi survey, the mean familiarity score (Cs), the mean judgment criteria score (Ca), and the mean authority coefficient (Cr) of the experts were all greater than 0.70 (Tables 8 and 9). The coordination coefficient (W) of the experts' opinions was 0.195 in the first round and 0.250 in the second round. The $\chi^2$ test indicated that the coordination of expert opinions was significant (p < 0.05), suggesting that the experts' opinions were well-coordinated and the results are reliable (Table 10).

Table 8: Expert authority coefficient (Cr) in the first round

| Themes | Cs | Ca | Cr |
|---|---|---|---|
| Patient Basic Information | 0.79 | 0.86 | 0.82 |
| Comprehensive Pain Assessment | 0.87 | 0.87 | 0.87 |
| Previous Analgesic Treatment | 0.83 | 0.80 | 0.81 |
| Evaluation of Previous Analgesic Treatment | 0.76 | 0.83 | 0.79 |
| Cancer Pain Medication Decision | 0.76 | 0.85 | 0.80 |
| Follow-up | 0.88 | 0.93 | 0.90 |

Table 9: Expert authority coefficient (Cr) in the second round

| Themes | Cs | Ca | Cr |
|---|---|---|---|
| Patient Basic Information | 0.83 | 0.88 | 0.85 |
| Comprehensive pain assessment | 0.87 | 0.86 | 0.86 |
| Previous analgesic treatment | 0.81 | 0.78 | 0.79 |
| Monitoring and management of analgesic-related adverse reactions | 0.87 | 0.90 | 0.88 |
| Drug accessibility | 0.77 | 0.82 | 0.79 |
| Lifestyle of patients | 0.91 | 0.80 | 0.85 |

Table 10: Coefficient of concordance (W) of experts in each round

| Delphi round | Items | W | $\chi^2$ | P |
|---|---|---|---|---|
| Round 1 | 21 | 0.195 | 126.779 | <0.001 |
| Round 2 | 7 | 0.250 | 54.163 | 0.006 |

As shown in Figure 4, consensus was reached on 20 feature items over two rounds of the Delphi process. From these 20 items, a total of 103 sub-items were included as features, covering six areas: basic patient information, comprehensive pain assessment, previous analgesic treatment and evaluation, cancer pain medication decision-making, monitoring and management of adverse reactions, and pain relief assessment.

**B.5 Feature Description**

Patients in the PEACE dataset have the following features (for data type, B: Binary, N: Numeric, M: Multiclass, *: Label):
**Patient Basic Information(50)**

1. **Demographics**
   - **ID (N)**: A unique random identification number assigned to each patient.
   - **Gender (B)**: The gender of the patient.
   - **Age (N)**: The age of the patient.
   - **Height (N)**: The height of the patient.
   - **Weight (N)**: The weight of the patient.
   - **BMI (N)**: A common indicator for assessing body fat, calculated using weight and height.

- **Body Surface Area (BSA) (N)**: The total surface area of the human body.
- **Medical Record Date (N)**: The date on which the doctor makes a decision regarding cancer pain medication treatment based on a comprehensive pain assessment.
- **Length of Hospital Stay (N)**: The duration of the patient's stay during the current hospital visit, measured in days.
- **Number of Hospital Admissions (N)**: The total number of times the patient has been hospitalized due to tumour diseases.
- **Diagnosis (M)**: The diagnosis provided by the doctor at the time of discharge, only including tumour-related diseases.
- **Smoking History (B)**: Whether the patient has a history of smoking continuously for 6 months or more.
- **Drinking History (B)**: Whether the patient has a history of drinking alcohol at least once a week for 6 months or more.
- **Allergy History (B)**: Whether the patient has experienced allergic reactions.
- **Tumour Treatment Methods (M)**: The methods of tumour treatment, including surgery, chemotherapy, radiotherapy, targeted therapy, and immunotherapy.
- **Gastrointestinal Risk (B)**: The likelihood of the patient developing gastrointestinal diseases (such as gastric ulcers, gastritis, enteritis) or related adverse reactions (such as gastrointestinal bleeding, indigestion) after taking pain medication.
- **Cardiovascular Risk (B)**: The likelihood of the patient developing cardiovascular diseases (such as hypertension, coronary heart disease, myocardial infarction) or related adverse reactions (such as arrhythmia, heart failure) after taking pain medication.
- **PS Score (N)**: The performance status score.

2. **Laboratory Examination Variables**

   (a) **Complete Blood Count:**
      - **White Blood Cell Count (N)**: The number of white blood cells in a unit volume of blood.
      - **Red Blood Cell Count (N)**: The number of red blood cells in a unit volume of blood.
      - **Hemoglobin (N)**: The amount of hemoglobin in a unit volume of blood.
      - **Platelet Count (N)**: The number of platelets in a unit volume of blood.
      - **Hematocrit (N)**: The volume percentage of red blood cells in blood.
      - **Neutrophil Count (N)**: The number of neutrophils in a unit volume of blood.
      - **Lymphocyte Count (N)**: The number of lymphocytes in a unit volume of blood.
      - **Eosinophil Count (N)**: The number of eosinophils in a unit volume of blood.
      - **Basophil Count (N)**: The number of basophils in a unit volume of blood.
      - **Monocyte Percentage (N)**: The proportion of monocytes in the total white blood cell count.
      - **Neutrophil Percentage (N)**: The proportion of neutrophils in the total white blood cell count.
      - **Lymphocyte Percentage (N)**: The proportion of lymphocytes in the total white blood cell count.
      - **Basophil Percentage (N)**: The proportion of basophils in the total white blood cell count.
      - **Eosinophil Percentage (N)**: The proportion of eosinophils in the total white blood cell count.
      - **Mean Corpuscular Volume (N)**: The average volume of a single red blood cell.
      - **Mean Corpuscular Hemoglobin (N)**: The average amount of hemoglobin in a single red blood cell.
      - **Mean Corpuscular Hemoglobin Concentration (N)**: The average concentration of hemoglobin in a single red blood cell.

- **Red Cell Distribution Width (N)**: The variation in the size of red blood cells.
- **Plateletcrit (N)**: The volume percentage of platelets in blood.
- **Mean Platelet Volume (N)**: The average volume of a single platelet.

(b) **Liver Function:**
- **Total Protein (N)**: The total amount of proteins in a unit volume of blood.
- **Albumin (N)**: The amount of albumin in a unit volume of blood.
- **Globulin (N)**: The amount of globulin in a unit volume of blood.
- **Albumin/Globulin Ratio (N)**: The ratio of albumin to globulin in blood.
- **Total Bilirubin (N)**: The total amount of bilirubin in a unit volume of blood.
- **Direct Bilirubin (N)**: The amount of direct (conjugated) bilirubin in a unit volume of blood.
- **Total Bile Acids (N)**: The total amount of bile acids in a unit volume of blood.
- **Alanine Aminotransferase (N)**: The amount of alanine aminotransferase (ALT) in a unit volume of blood.
- **Aspartate Aminotransferase (N)**: The amount of aspartate aminotransferase (AST) in a unit volume of blood.

(c) **Kidney Function:**
- **Urea (N)**: The amount of urea in a unit volume of blood, reflecting kidney excretory function.
- **Creatinine (N)**: The amount of creatinine in a unit volume of blood, reflecting kidney filtration function.
- **Uric Acid (N)**: The amount of uric acid in a unit volume of blood, reflecting kidney excretory function and purine metabolism status.

**Comprehensive Pain Assessment (15):**

- **Pain Type (M)**: Classification of pain based on the pathological mechanism.
- **Worst Pain (N)**: The highest level of pain experienced in the last 24 hours, assessed using the Numerical Rating Scale (NRS).
- **Mildest Pain (N)**: The lowest level of pain experienced in the last 24 hours, assessed using NRS.
- **Average Pain (N)**: The average level of pain experienced in the last 24 hours, assessed using NRS.
- **Current Pain (N)**: The current level of pain, assessed using NRS.
- **Impact of Pain on Daily Life (N)**: The degree to which daily life was affected by pain in the past week.
- **Impact of Pain on Mood (N)**: The degree to which mood was affected by pain in the past week.
- **Impact of Pain on Walking Ability (N)**: The degree to which walking ability was affected by pain in the past week.
- **Impact of Pain on Daily Work (N)**: The degree to which daily work was affected by pain in the past week.
- **Impact of Pain on Relationships with Others (N)**: The degree to which relationships with others were affected by pain in the past week.
- **Impact of Pain on Sleep (N)**: The degree to which sleep was affected by pain in the past week.
- **Impact of Pain on Interest in Life (N)**: The degree to which interest in life was affected by pain in the past week.
- **Pain Frequency (M)**: The number of times pain occurred in a day for cancer pain patients.

- **Type of Breakthrough Pain (M)**: Classification of breakthrough pain according to the National Comprehensive Cancer Network (NCCN).
- **Frequency of Breakthrough Pain (M)**: The number of times breakthrough pain occurred in a day for cancer pain patients.

**Previous Analgesic Treatment(23):**

- **Prev_Extended Release Strong Opiates (ERSO) (N)**: The number of types of extended-release strong opiates used by the patient in the past week.
- **Prev_Immediate Release Strong Opiates (IRSO) (N)**: The number of types of immediate-release strong opiates used by the patient in the past week.
- **Prev_Extended Release Weak Opiates (ERWO) (N)**: The number of types of extended-release weak opiates used by the patient in the past week.
- **Prev_Immediate Release Weak Opiates (IRWO) (N)**: The number of types of immediate-release weak opiates used by the patient in the past week.
- **Prev_Nonsteroidal Anti-inflammatory Drugs (NSAID) (N)**: The number of types of nonsteroidal anti-inflammatory drugs used by the patient in the past week.
- **Prev_Anticonvulsants/Antidepressants (A/A) (N)**: The number of types of anticonvulsants/antidepressants used by the patient in the past week.
- **Prev_Others (N)**: The number of other analgesics used by the patient in the past week, excluding ERSO, IRSO, ERWO, IRWO, NSAIDs, and A/A.
- **Opiate Tolerance (B)**: Whether the patient has developed a decreased effect or reduced duration of action when using opiates for pain treatment.
- **Days of Medication Use (N)**: The number of days the patient used opiates (calculated based on the highest level of opiates used if multiple types were used simultaneously).
- The following 9 items are from the Morisky Medication Adherence Scale (MMAS-8), including 8 questions and a total score:
  - **M1 (N)**: Do you sometimes forget to take your medications?
  - **M2 (N)**: People sometimes miss taking their medications for reasons other than forgetting. Thinking over the past two weeks, were there any days when you did not take your medications?
  - **M3 (N)**: Have you ever cut back or stopped taking your medications without telling your doctor because you felt worse when you took them?
  - **M4 (N)**: When you travel or leave home, do you sometimes forget to bring along your medications?
  - **M5 (N)**: Did you take all your medications yesterday?
  - **M6 (N)**: When you feel like your symptoms are under control, do you sometimes stop taking your medications?
  - **M7 (N)**: Taking medication every day is a real inconvenience for some people. Do you ever feel hassled about sticking to your treatment plan?
  - **M8 (N)**: Do you have difficulty remembering to take all your medications?
  - **MMAS-8 Total Score (N)**: The total score ranges from M1 to M8, with higher scores indicating better adherence to medication.
- **Duration of Analgesic Control (N)**: The duration of pain control after taking analgesics.
- **Constipation (B)**: Whether the patient experienced constipation as an adverse reaction after taking analgesics.
- **Nausea/Vomiting (B)**: Whether the patient experienced nausea or vomiting as an adverse reaction after taking analgesics.

- **Other Adverse Reactions (B)**: Whether the patient experienced other adverse reactions besides constipation and nausea/vomiting after taking analgesics.
- **Medication for Adverse Reactions (B)**: Whether the patient used medications to manage adverse reactions.

**Evaluation of Previous Analgesic Treatment(5):**

1. The following 5 features are classified according to the Pharmaceutical Care Network Europe (PCNE) V8.0 classification of drug-related problems (DRPs):
   - **Drug-Related Problems (DRPs) (M)**: Any undesirable outcome or potential issue arising during the patient's drug therapy. This includes aspects of treatment effectiveness and safety.
   - **Causes (M)**: The underlying causes or factors leading to drug therapy problems.
   - **Interventions (M)**: Specific actions or measures taken to address drug therapy problems. These interventions can be implemented by pharmacists, doctors, or other healthcare professionals.
   - **Acceptance of Interventions (M)**: The patient's acceptance of the intervention plans proposed by healthcare professionals.
   - **Status of DRPs (M)**: The resolution status of DRPs after healthcare professionals' intervention.

**Cancer Pain Medication Decision(9):**

- **ERSO_Recommended (N*)**: The number of extended-release strong opiates recommended by the doctor.
- **IRSO_Recommended (N*)**: The number of immediate-release strong opiates recommended by the doctor.
- **ERWO_Recommended (N*)**: The number of extended-release weak opiates recommended by the doctor.
- **IRWO_Recommended (N*)**: The number of immediate-release weak opiates recommended by the doctor.
- **NSAIDs_Recommended (N*)**: The number of nonsteroidal anti-inflammatory drugs recommended by the doctor.
- **A/A_Recommended (N*)**: The number of anticonvulsants/antidepressants recommended by the doctor.
- **Others (N*)**: The number of other analgesics recommended by the doctor, excluding ERSO, IRSO, ERWO, IRWO, NSAIDs, and A/A.
- **Constipation Medication Recommended (M)**: The types of medication recommended by the doctor for managing constipation.
- **Nausea/Vomiting Medication Recommended (M)**: The types of medication recommended by the doctor for managing nausea and vomiting.

**Follow-up(1):**

- **Pain Relief Status (M*)**: The degree of pain relief experienced by the patient after using the analgesic regimen recommended by the doctor.

# C   Demographics

This section examines the age distribution within the PEACE dataset. We analyze the population breakdown across different age groups, as detailed in Table 11. The table categorizes the number of individuals in each age group by gender.

Table 11: Population Distribution

| Age Group | Number | Male | Female |
|-----------|--------|------|--------|
| 18-29 | 2,681 | 1,931 | 750 |
| 30-44 | 7,675 | 5,045 | 2,630 |
| 45-59 | 14,737 | 7,663 | 7,074 |
| 60-74 | 11,054 | 4,316 | 6,738 |
| $\geq 75$ | 2,619 | 969 | 1,650 |
| Total | 38,766 | 18,842 | 19,924 |

# D  Training Details

## D.1  Baseline Models

The source code of the models used in our experiments is available at https://github.com/YTYTYD/PEACE/tree/main/Code.

**Basic machine learning and neural network models:**

1. Decision Trees[22]: A machine learning algorithm that predicts outcomes by recursively splitting data into subsets based on feature values, forming a tree structure of decisions.

2. Logistic Regression[5]: A machine learning algorithm used for both classification and regression tasks that models the probability of outcomes using a logistic function.

3. Random Forests[14]: A machine learning algorithm that employs an ensemble of decision trees to improve prediction accuracy and control overfitting by aggregating the predictions of multiple trees.

4. Support Vector Machines (SVM)[4]: A machine learning algorithm for classification and regression that identifies the optimal hyperplane to separate different classes in a high-dimensional space.

5. Multilayer Perceptrons (MLP)[23]: A neural network algorithm composed of multiple layers of neurons, capable of performing various tasks including classification and regression.

**Gradient boosting decision tree models:**

1. LightGBM[12]: is an advanced machine learning algorithm that implements gradient boosting on decision trees using a leaf-wise growth strategy, offering superior performance and computational efficiency for large-scale and high-dimensional datasets.

2. XGBoost[3]: is a highly optimised and scalable machine learning algorithm that applies gradient boosting techniques with features like regularisation, parallel processing, and tree pruning, achieving exceptional performance and accuracy in various predictive modelling tasks.

3. AdaBoost[6]: is a machine learning algorithm that enhances classification and regression accuracy by iteratively combining multiple weak classifiers into a strong classifier, focusing on misclassified instances to improve overall model performance.

**Advanced neural network models:**

1. iTransformer[15]: is a neural network algorithm specifically designed for time series forecasting. It inverts the traditional transformer architecture to better capture temporal dependencies and sequence patterns in time series data. By reversing the order of attention mechanisms, iTransformer focuses on leveraging past data more effectively to predict future values. The algorithm employs a novel architecture that integrates both local and global temporal information, leading to significant improvements in forecasting accuracy.

2. Transtab[29]: is a neural network algorithm based on transformer architecture, designed to handle tabular data with varying structures by converting each row into a generalisable embedding vector and using stacked transformers for feature encoding. It combines column descriptions and table cells as input to a gated transformer model and leverages supervised and self-supervised pretraining to enhance performance. Transtab excels in learning from multiple tables with partially overlapping columns and updating models incrementally, achieving top rankings in supervised, incremental, and transfer learning tasks across diverse datasets.

3. Mamba[8]: is a neural network algorithm that addresses the inefficiencies of transformer models in sequence modeling. By using selective state space models (SSMs) where parameters depend on the input, Mamba can selectively retain or discard information, achieving linear scaling in sequence length without attention or MLP blocks. This design enables faster inference and high throughput, demonstrating state-of-the-art performance across various domains, including language, audio, and genomics, and outperforming similarly sized transformers.

**EHR-specific models:**

1. Stagenet[7]: is a neural network model designed for health risk prediction, leveraging the identification of different stages in a patient's disease progression to improve prediction accuracy. The model consists of two key modules: the stage-aware LSTM module, which automatically and unsupervisedly extracts stage variations in a patient's health condition, and the stage-adaptive convolutional module, which uses convolution operations to capture health progression patterns from these stages, focusing on stage-specific features and recalibrating them to enhance prediction outcomes.

2. Adacare[16]: is a health status representation learning model focused on EHR, capable of capturing the variation trends of biomarkers in both long-term and short-term scales. It uses dilated convolutions to capture features across different time scales. Additionally, it incorporates a scale-adaptive feature recalibration module, which adaptively enhances important features based on the patient's health condition while suppressing irrelevant features.

## D.2 Data splitting

Data splitting for model training. see Figure 5. For the TEA task, we removed some records with missing labels.

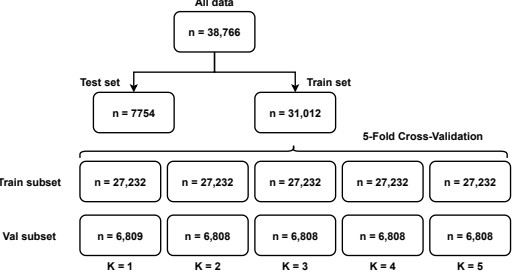

Figure 5: Data splitting for PEACE dataset

## D.3 Evaluation Metrics

This section describes the metrics used to evaluate the performance of the trained model. For classification tasks, TP (True Positive) is a true positive, TN (True Negative) is a true negative, FP (False Positive) is a false positive, and FN (False Negative) is a false negative. Our evaluation metrics

and calculation methods are shown in Table 12. For regression tasks, $y_i$ is the actual value, $\hat{y}_i$ is the predicted value, and $n$ is the number of observations. Our evaluation metrics and calculation methods are shown in Table 13.

Table 12: Classification evaluation metrics

| Metric | Explanation and Formula |
|---|---|
| Accuracy (ACC) | **Explanation:** Accuracy is the proportion of correctly predicted samples out of the total samples. 
 **Formula:** Accuracy $= \frac{TP+TN}{TP+TN+FP+FN}$ |
| Area Under the Receiver Operating Characteristic Curve (AUROC) | **Explanation:** AUROC is the area under the ROC curve, which evaluates the performance of a classification model. The ROC curve shows the trade-off between the true positive rate (TPR) and false positive rate (FPR) at various threshold settings. |
| Recall | **Explanation:** Recall is the proportion of true positives correctly identified by the model out of all actual positives. 
 **Formula:** Recall $= \frac{TP}{TP+FN}$ |
| Precision | **Explanation:** Precision is the proportion of true positives correctly identified by the model out of all predicted positives. 
 **Formula:** Precision $= \frac{TP}{TP+FP}$ |
| F1 Score | **Explanation:** The F1 score is the harmonic mean of precision and recall, providing a balance between the two. 
 **Formula:** $F1 = 2 \times \frac{\text{Precision} \times \text{Recall}}{\text{Precision} + \text{Recall}}$ |

Table 13: Regression evaluation metrics

| Metric | Explanation and Formula |
|---|---|
| Mean Squared Error (MSE) | **Explanation:** MSE measures the average squared difference between the predicted values and the actual values. It gives a higher weight to larger errors, making it sensitive to outliers. 
 **Formula:** MSE $= \frac{1}{n} \sum_{i=1}^{n}(y_i - \hat{y}_i)^2$ |
| Mean Absolute Error (MAE) | **Explanation:** MAE measures the average absolute difference between the predicted values and the actual values. It gives equal weight to all errors, making it less sensitive to outliers. 
 **Formula:** MAE $= \frac{1}{n} \sum_{i=1}^{n} |y_i - \hat{y}_i|$ |

## D.4 Detailed Experimental Results

Tables 14 and 15 respectively present the performance evaluation details of TEA and MR tasks, including the detailed evaluation metrics for each fold, the mean and error of the 5-folds, and the values for the independent test set. The statistical and analytical processing of experimental results retains four decimal places to minimise rounding errors. We acknowledge that data processing and visualisation tasks, including calculations of means and errors, are supported by large language models (LLMs).

# E Release and Usage of Dataset

We release the PEACE dataset under a CC-BY license. The dataset access involves three steps:

1. Complete some training and provide certification (such as the CITI or GCP certification).

2. Carefully read the terms of the Data Use Agreement and if you agree and wish to proceed, send your application to the manager. Please use an official email address (such as .edu).

3. Final approval of data access is required by Xiangya Hospital

Once an application is approved, the researcher will receive an email with instructions for downloading the dataset. We estimate a response time of 20 business days for processing requests. This duration may vary depending on the completeness of the provided information and can take up to three months. Any model trained on this dataset should not be deployed in real-world systems

Table 14: Details of TEA Task Model Performance Evaluation

**Decision Tree**

| Metric | Fold 1 | Fold 2 | Fold 3 | Fold 4 | Fold 5 | Mean ± SE | Test |
|---|---|---|---|---|---|---|---|
| Accuracy | 0.7105 | 0.7297 | 0.7178 | 0.7224 | 0.7139 | 0.7189±0.0030 | 0.7236 |
| F1 Score | 0.6588 | 0.6770 | 0.6531 | 0.6630 | 0.6590 | 0.6622±0.0035 | 0.6659 |
| Recall | 0.6640 | 0.6797 | 0.6574 | 0.6627 | 0.6590 | 0.6645±0.0037 | 0.6745 |
| Precision | 0.6540 | 0.6747 | 0.6490 | 0.6633 | 0.6593 | 0.6601±0.0042 | 0.6591 |
| AUROC | 0.7758 | 0.7876 | 0.7744 | 0.7775 | 0.7738 | 0.7778±0.0025 | 0.7838 |

**Logistic Regression**

| Metric | Fold 1 | Fold 2 | Fold 3 | Fold 4 | Fold 5 | Mean ± SE | Test |
|---|---|---|---|---|---|---|---|
| Accuracy | 0.6743 | 0.6752 | 0.6822 | 0.6729 | 0.6854 | 0.6780±0.0022 | 0.6836 |
| F1 Score | 0.6040 | 0.6079 | 0.5989 | 0.6025 | 0.6065 | 0.6040±0.0015 | 0.6028 |
| Recall | 0.5740 | 0.5795 | 0.5694 | 0.5754 | 0.5707 | 0.5738±0.0018 | 0.5730 |
| Precision | 0.6748 | 0.6760 | 0.6713 | 0.6671 | 0.6988 | 0.6776±0.0052 | 0.6734 |
| AUROC | 0.7198 | 0.7225 | 0.7190 | 0.7200 | 0.7198 | 0.7202±0.0011 | 0.7204 |

**Random Forest**

| Metric | Fold 1 | Fold 2 | Fold 3 | Fold 4 | Fold 5 | Mean ± SE | Test |
|---|---|---|---|---|---|---|---|
| Accuracy | 0.7800 | 0.7830 | 0.7855 | 0.7886 | 0.7858 | 0.7846±0.0014 | 0.7916 |
| F1 Score | 0.7405 | 0.7396 | 0.7318 | 0.7370 | 0.7382 | 0.7374±0.0016 | 0.7416 |
| Recall | 0.7044 | 0.7025 | 0.6936 | 0.7006 | 0.6994 | 0.7001±0.0020 | 0.7031 |
| Precision | 0.8119 | 0.8123 | 0.8082 | 0.8089 | 0.8125 | 0.8108±0.0010 | 0.8139 |
| AUROC | 0.8084 | 0.8079 | 0.8041 | 0.8083 | 0.8069 | 0.8071±0.0010 | 0.8097 |

**SVM**

| Metric | Fold 1 | Fold 2 | Fold 3 | Fold 4 | Fold 5 | Mean ± SE | Test |
|---|---|---|---|---|---|---|---|
| Accuracy | 0.6598 | 0.6580 | 0.6691 | 0.6634 | 0.6735 | 0.6648±0.0025 | 0.6694 |
| F1 Score | 0.5679 | 0.5627 | 0.5680 | 0.5771 | 0.5658 | 0.5683±0.0024 | 0.5678 |
| Recall | 0.5474 | 0.5478 | 0.5434 | 0.5542 | 0.5396 | 0.5465±0.0025 | 0.5459 |
| Precision | 0.6533 | 0.6433 | 0.6612 | 0.6555 | 0.6789 | 0.6584±0.0058 | 0.6533 |
| AUROC | 0.7023 | 0.7018 | 0.7018 | 0.7063 | 0.7009 | 0.7026±0.0011 | 0.7028 |

**MLP**

| Metric | Fold 1 | Fold 2 | Fold 3 | Fold 4 | Fold 5 | Mean ± SE | Test |
|---|---|---|---|---|---|---|---|
| Accuracy | 0.7257 | 0.7347 | 0.7402 | 0.7399 | 0.7464 | 0.7374±0.0033 | 0.7508 |
| F1 Score | 0.6746 | 0.6859 | 0.6738 | 0.6807 | 0.6857 | 0.6801±0.0023 | 0.6955 |
| Recall | 0.6673 | 0.6787 | 0.6720 | 0.6781 | 0.6691 | 0.6730±0.0020 | 0.6831 |
| Precision | 0.6852 | 0.6941 | 0.6771 | 0.6840 | 0.7076 | 0.6896±0.0048 | 0.7109 |
| AUROC | 0.7799 | 0.7874 | 0.7853 | 0.7885 | 0.7848 | 0.7852±0.0015 | 0.7925 |

**XGBoost**

| Metric | Fold 1 | Fold 2 | Fold 3 | Fold 4 | Fold 5 | Mean ± SE | Test |
|---|---|---|---|---|---|---|---|
| Accuracy | 0.7840 | 0.7976 | 0.7966 | 0.7989 | 0.7963 | 0.7947±0.0023 | 0.8063 |
| F1 Score | 0.7501 | 0.7579 | 0.7434 | 0.7532 | 0.7474 | 0.7504±0.0024 | 0.7607 |
| Recall | 0.7218 | 0.7261 | 0.7114 | 0.7234 | 0.7101 | 0.7186±0.0033 | 0.7301 |
| Precision | 0.7952 | 0.8080 | 0.7964 | 0.8012 | 0.8107 | 0.8023±0.0027 | 0.8080 |
| AUROC | 0.8182 | 0.8226 | 0.8155 | 0.8220 | 0.8144 | 0.8185±0.0015 | 0.8265 |

**LightGBM**

| Metric | Fold 1 | Fold 2 | Fold 3 | Fold 4 | Fold 5 | Mean ± SE | Test |
|---|---|---|---|---|---|---|---|
| Accuracy | 0.7925 | 0.8067 | 0.8053 | 0.8038 | 0.8034 | 0.8023±0.0024 | 0.8108 |
| F1 Score | 0.7623 | 0.7717 | 0.7569 | 0.7577 | 0.7592 | 0.7616±0.0023 | 0.7723 |
| Recall | 0.7338 | 0.7392 | 0.7233 | 0.7282 | 0.7240 | 0.7297±0.0028 | 0.7406 |
| Precision | 0.8093 | 0.8258 | 0.8149 | 0.8079 | 0.8195 | 0.8155±0.0031 | 0.8234 |
| AUROC | 0.8261 | 0.8315 | 0.8232 | 0.8257 | 0.8231 | 0.8259±0.0015 | 0.8327 |

**AdaBoost**

| Metric | Fold 1 | Fold 2 | Fold 3 | Fold 4 | Fold 5 | Mean ± SE | Test |
|---|---|---|---|---|---|---|---|
| Accuracy | 0.6620 | 0.6671 | 0.6614 | 0.6684 | 0.6646 | 0.6647±0.0011 | 0.6852 |
| F1 Score | 0.5668 | 0.5534 | 0.5614 | 0.5682 | 0.5480 | 0.5596±0.0031 | 0.5961 |
| Recall | 0.5567 | 0.5527 | 0.5470 | 0.5622 | 0.5377 | 0.5513±0.0038 | 0.5725 |
| Precision | 0.6395 | 0.6069 | 0.6372 | 0.6320 | 0.6451 | 0.6321±0.0073 | 0.6776 |
| AUROC | 0.7063 | 0.7056 | 0.7010 | 0.7114 | 0.6972 | 0.7043±0.0027 | 0.7196 |

**Transtab**

| Metric | Fold 1 | Fold 2 | Fold 3 | Fold 4 | Fold 5 | Mean ± SE | Test |
|---|---|---|---|---|---|---|---|
| Accuracy | 0.5840 | 0.5716 | 0.5922 | 0.5877 | 0.5822 | 0.5835±0.0034 | 0.5835 |
| F1 Score | 0.3129 | 0.3087 | 0.3246 | 0.3169 | 0.3217 | 0.3170±0.0029 | 0.3170 |
| Recall | 0.3394 | 0.3390 | 0.3532 | 0.3470 | 0.3459 | 0.3449±0.0026 | 0.3449 |
| Precision | 0.3524 | 0.2834 | 0.4815 | 0.2918 | 0.3305 | 0.3479±0.0357 | 0.3479 |
| AUROC | 0.6623 | 0.6562 | 0.6627 | 0.6656 | 0.6505 | 0.6595±0.0027 | 0.6594 |

**iTransformer**

| Metric | Fold 1 | Fold 2 | Fold 3 | Fold 4 | Fold 5 | Mean ± SE | Test |
|---|---|---|---|---|---|---|---|
| Accuracy | 0.5762 | 0.7325 | 0.6918 | 0.5573 | 0.7473 | 0.6606±0.0396 | 0.6831 |
| F1 Score | 0.5497 | 0.7269 | 0.6872 | 0.5328 | 0.7348 | 0.6456±0.0437 | 0.6827 |
| Recall | 0.5765 | 0.7332 | 0.6915 | 0.5576 | 0.7476 | 0.6608±0.0397 | 0.6839 |
| Precision | 0.5496 | 0.7248 | 0.6965 | 0.5524 | 0.7327 | 0.6506±0.0413 | 0.6817 |
| AUROC | 0.6583 | 0.7405 | 0.7266 | 0.6517 | 0.7433 | 0.7036±0.0203 | 0.7340 |

**Mamba**

| Metric | Fold 1 | Fold 2 | Fold 3 | Fold 4 | Fold 5 | Mean ± SE | Test |
|---|---|---|---|---|---|---|---|
| Accuracy | 0.6723 | 0.7608 | 0.7364 | 0.6577 | 0.8091 | 0.7272±0.0281 | 0.7606 |
| F1 Score | 0.6545 | 0.7623 | 0.7406 | 0.6414 | 0.8094 | 0.7212±0.0322 | 0.7625 |
| Recall | 0.6728 | 0.7609 | 0.7357 | 0.6579 | 0.8102 | 0.7272±0.0282 | 0.7630 |
| Precision | 0.6802 | 0.7633 | 0.7511 | 0.6731 | 0.8094 | 0.7352±0.0259 | 0.7621 |
| AUROC | 0.7315 | 0.7902 | 0.7813 | 0.7164 | 0.8256 | 0.7686±0.0200 | 0.7959 |

**StageNet**

| Metric | Fold 1 | Fold 2 | Fold 3 | Fold 4 | Fold 5 | Mean ± SE | Test |
|---|---|---|---|---|---|---|---|
| Accuracy | 0.6790 | 0.7792 | 0.7041 | 0.7291 | 0.7541 | 0.7291 ± 0.0177 | 0.7832 |
| F1 Score | 0.7422 | 0.8361 | 0.7659 | 0.7894 | 0.8128 | 0.7893 ± 0.0166 | 0.7271 |
| Recall | 0.6880 | 0.7536 | 0.7044 | 0.7208 | 0.7372 | 0.7208 ± 0.0116 | 0.6898 |
| Precision | 0.8059 | 0.9391 | 0.8392 | 0.8725 | 0.9058 | 0.8725 ± 0.0235 | 0.7688 |
| AUROC | 0.7349 | 0.7537 | 0.7396 | 0.7443 | 0.7490 | 0.7443 ± 0.0033 | 0.7443 |

**AdaCare**

| Metric | Fold 1 | Fold 2 | Fold 3 | Fold 4 | Fold 5 | Mean ± SE | Test |
|---|---|---|---|---|---|---|---|
| Accuracy | 0.7106 | 0.7598 | 0.755 | 0.7681 | 0.7599 | 0.7507 ± 0.0111 | 0.7582 |
| F1 Score | 0.6664 | 0.7185 | 0.7203 | 0.7251 | 0.7136 | 0.7087 ± 0.0107 | 0.7252 |
| Recall | 0.5964 | 0.6746 | 0.6794 | 0.6834 | 0.6720 | 0.6612 ± 0.0326 | 0.6836 |
| Precision | 0.7550 | 0.7684 | 0.7664 | 0.7724 | 0.7609 | 0.7646 ± 0.0061 | 0.7724 |
| AUROC | 0.8438 | 0.8511 | 0.8546 | 0.8578 | 0.8499 | 0.8515 ± 0.0047 | 0.8588 |

## Table 15: Details of MR Task Model Performance Evaluation

### ERSO

| Model | Fold 1 MSE | Fold 2 MSE | Fold 3 MSE | Fold 4 MSE | Fold 5 MSE | Mean MSE | Test MSE | Fold 1 MAE | Fold 2 MAE | Fold 3 MAE | Fold 4 MAE | Fold 5 MAE | Mean MAE | Test MAE |
|---|---|---|---|---|---|---|---|---|---|---|---|---|---|---|
| Decision Tree | 0.0333 | 0.0363 | 0.0373 | 0.0401 | 0.0394 | 0.0373±0.0012 | 0.0473 | 0.0322 | 0.0354 | 0.0358 | 0.0389 | 0.0376 | 0.0360±0.0011 | 0.0443 |
| Logistic Regression | 0.1262 | 0.1295 | 0.1234 | 0.1289 | 0.1311 | 0.1278±0.0014 | 0.1328 | 0.1259 | 0.1292 | 0.1228 | 0.1281 | 0.1297 | 0.1271±0.0013 | 0.1321 |
| Random Forest | 0.0172 | 0.0184 | 0.0189 | 0.0199 | 0.0202 | 0.0189±0.0005 | 0.0234 | 0.0357 | 0.0364 | 0.0390 | 0.0391 | 0.0396 | 0.0380±0.0008 | 0.0440 |
| SVM | 0.1545 | 0.1630 | 0.1672 | 0.1951 | 0.1721 | 0.1704±0.0068 | 0.3417 | 0.2701 | 0.2677 | 0.2704 | 0.2719 | 0.2810 | 0.2722±0.0023 | 0.2764 |
| MLP | 0.0378 | 0.0337 | 0.0396 | 0.0357 | 0.0386 | 0.0371±0.0016 | 0.6994 | 0.1125 | 0.1120 | 0.1140 | 0.1140 | 0.1221 | 0.1149±0.0018 | 0.1219 |
| XGBoost | 0.0190 | 0.0206 | 0.0210 | 0.0230 | 0.0213 | 0.0210±0.0006 | 0.0265 | 0.0493 | 0.0523 | 0.0551 | 0.0554 | 0.0510 | 0.0526±0.0012 | 0.0572 |
| LightGBM | 0.0174 | 0.0186 | 0.0192 | 0.0198 | 0.0197 | 0.0189±0.0004 | 0.0236 | 0.0437 | 0.0435 | 0.0454 | 0.0454 | 0.0455 | 0.0447±0.0005 | 0.0497 |
| AdaBoost | 0.1697 | 0.2267 | 0.1774 | 0.1952 | 0.1875 | 0.1913±0.0099 | 0.1947 | 0.3947 | 0.4734 | 0.4012 | 0.4311 | 0.4233 | 0.4247±0.0139 | 0.4293 |
| Transtab | 0.2796 | 0.2857 | 0.2804 | 0.2831 | 0.2850 | 0.2828±0.0012 | 0.2828 | 0.2787 | 0.2855 | 0.2785 | 0.2828 | 0.2833 | 0.2818±0.0014 | 0.2818 |
| iTransfomer | 0.0573 | 0.0181 | 0.0264 | 0.0911 | 0.0285 | 0.0442±0.0134 | 0.0259 | 0.0762 | 0.0716 | 0.0815 | 0.1423 | 0.0875 | 0.0808±0.0152 | 0.0257 |
| Mamba | 0.0214 | 0.0193 | 0.0249 | 0.0780 | 0.0130 | 0.0313±0.0118 | 0.0190 | 0.0388 | 0.0431 | 0.0590 | 0.1178 | 0.0380 | 0.0526±0.0140 | 0.0387 |
| StageNet | 0.0342 | 0.0210 | 0.0327 | 0.0416 | 0.0188 | 0.0297±0.0042 | 0.0288 | 0.0775 | 0.0717 | 0.0772 | 0.082 | 0.0699 | 0.0756±0.0021 | 0.0736 |
| AdaCare | 0.0254 | 0.0179 | 0.0295 | 0.0327 | 0.0175 | 0.0246±0.0031 | 0.0221 | 0.0271 | 0.0225 | 0.0339 | 0.0347 | 0.0223 | 0.0281±0.0027 | 0.0261 |

### IRSO

| Model | Fold 1 MSE | Fold 2 MSE | Fold 3 MSE | Fold 4 MSE | Fold 5 MSE | Mean MSE | Test MSE | Fold 1 MAE | Fold 2 MAE | Fold 3 MAE | Fold 4 MAE | Fold 5 MAE | Mean MAE | Test MAE |
|---|---|---|---|---|---|---|---|---|---|---|---|---|---|---|
| Decision Tree | 0.0269 | 0.0263 | 0.0336 | 0.0305 | 0.0273 | 0.0289±0.0014 | 0.0309 | 0.0269 | 0.0263 | 0.0333 | 0.0305 | 0.0273 | 0.0289±0.0013 | 0.0306 |
| Logistic Regression | 0.1163 | 0.1079 | 0.1203 | 0.1126 | 0.1137 | 0.1142±0.0021 | 0.1114 | 0.1163 | 0.1079 | 0.1203 | 0.1126 | 0.1137 | 0.1142±0.0021 | 0.1107 |
| Random Forest | 0.0153 | 0.0134 | 0.0179 | 0.0153 | 0.0155 | 0.0155±0.0007 | 0.0158 | 0.0305 | 0.0279 | 0.0340 | 0.0318 | 0.0313 | 0.0311±0.0010 | 0.0322 |
| SVM | 0.1651 | 0.1577 | 0.1802 | 0.1682 | 0.1605 | 0.1663±0.0039 | 0.1713 | 0.1609 | 0.1561 | 0.1716 | 0.1624 | 0.1555 | 0.1613±0.0028 | 0.1619 |
| MLP | 0.0315 | 0.0296 | 0.0326 | 0.0332 | 0.0310 | 0.0316±0.0006 | 0.1076 | 0.1022 | 0.1047 | 0.1077 | 0.1105 | 0.1048 | 0.1060±0.0014 | 0.1050 |
| XGBoost | 0.0165 | 0.0143 | 0.0188 | 0.0169 | 0.0160 | 0.0165±0.0007 | 0.0164 | 0.0420 | 0.0392 | 0.0445 | 0.0424 | 0.0421 | 0.0420±0.0008 | 0.0430 |
| LightGBM | 0.0150 | 0.0137 | 0.0179 | 0.0156 | 0.0147 | 0.0154±0.0007 | 0.0156 | 0.0357 | 0.0347 | 0.0396 | 0.0371 | 0.0363 | 0.0367±0.0008 | 0.0365 |
| AdaBoost | 0.0779 | 0.0491 | 0.0489 | 0.1262 | 0.0793 | 0.0763±0.0141 | 0.1083 | 0.2304 | 0.1485 | 0.1467 | 0.3335 | 0.2301 | 0.2178±0.0343 | 0.2940 |
| Transtab | 0.2358 | 0.2393 | 0.2295 | 0.2274 | 0.2332 | 0.2330±0.0021 | 0.2330 | 0.2358 | 0.2393 | 0.2290 | 0.2274 | 0.2330 | 0.2329±0.0022 | 0.2329 |
| iTransfomer | 0.0851 | 0.0399 | 0.0343 | 0.0936 | 0.0157 | 0.0537±0.0151 | 0.0375 | 0.1275 | 0.1192 | 0.1164 | 0.1820 | 0.0720 | 0.1091±0.0203 | 0.1181 |
| Mamba | 0.0223 | 0.0100 | 0.0112 | 0.0552 | 0.0083 | 0.0214±0.0088 | 0.0100 | 0.0246 | 0.0324 | 0.0283 | 0.1021 | 0.0264 | 0.0373±0.0133 | 0.0339 |
| StageNet | 0.1845 | 0.1769 | 0.1810 | 0.1857 | 0.1709 | 0.1798±0.0027 | 0.1810 | 0.3632 | 0.3556 | 0.3597 | 0.3644 | 0.3496 | 0.3585±0.0027 | 0.3590 |
| AdaCare | 0.0175 | 0.0152 | 0.0162 | 0.0191 | 0.0145 | 0.0165±0.0009 | 0.0156 | 0.0201 | 0.0186 | 0.0190 | 0.0218 | 0.0175 | 0.0194±0.0007 | 0.0185 |

### ERWO

| Model | Fold 1 MSE | Fold 2 MSE | Fold 3 MSE | Fold 4 MSE | Fold 5 MSE | Mean MSE | Test MSE | Fold 1 MAE | Fold 2 MAE | Fold 3 MAE | Fold 4 MAE | Fold 5 MAE | Mean MAE | Test MAE |
|---|---|---|---|---|---|---|---|---|---|---|---|---|---|---|
| Decision Tree | 0.0082 | 0.0112 | 0.0116 | 0.0100 | 0.0093 | 0.0101±0.0006 | 0.0130 | 0.0082 | 0.0112 | 0.0116 | 0.0100 | 0.0093 | 0.0101±0.0006 | 0.0130 |
| Logistic Regression | 0.0203 | 0.0216 | 0.0216 | 0.0181 | 0.0210 | 0.0205±0.0007 | 0.0221 | 0.0203 | 0.0216 | 0.0216 | 0.0181 | 0.0210 | 0.0205±0.0007 | 0.0221 |
| Random Forest | 0.0055 | 0.0061 | 0.0056 | 0.0054 | 0.0052 | 0.0056±0.0002 | 0.0069 | 0.0111 | 0.0119 | 0.0122 | 0.0108 | 0.0108 | 0.0114±0.0003 | 0.0137 |
| SVM | 0.0300 | 0.0301 | 0.0298 | 0.0332 | 0.0317 | 0.0310±0.0007 | 0.0319 | 0.0300 | 0.0301 | 0.0298 | 0.0332 | 0.0317 | 0.0310±0.0007 | 0.0319 |
| MLP | 0.0171 | 0.0109 | 0.0105 | 0.0104 | 0.0110 | 0.0120±0.0013 | 0.0691 | 0.0717 | 0.0511 | 0.0510 | 0.0505 | 0.0501 | 0.0661±0.0048 | 0.0518 |
| XGBoost | 0.0074 | 0.0066 | 0.0073 | 0.0064 | 0.0061 | 0.0068±0.0003 | 0.0079 | 0.0165 | 0.0166 | 0.0177 | 0.0159 | 0.0154 | 0.0164±0.0004 | 0.0191 |
| LightGBM | 0.0056 | 0.0061 | 0.0059 | 0.0053 | 0.0053 | 0.0056±0.0002 | 0.0065 | 0.0135 | 0.0141 | 0.0144 | 0.0124 | 0.0134 | 0.0136±0.0004 | 0.0151 |
| AdaBoost | 0.0193 | 0.0173 | 0.0206 | 0.0156 | 0.0096 | 0.0165±0.0019 | 0.0192 | 0.0486 | 0.0472 | 0.0569 | 0.0446 | 0.0186 | 0.0432±0.0065 | 0.0529 |
| Transtab | 0.0294 | 0.0309 | 0.0296 | 0.0314 | 0.0279 | 0.0298±0.0006 | 0.0298 | 0.0294 | 0.0309 | 0.0296 | 0.0314 | 0.0279 | 0.0302±0.0004 | 0.0298 |
| iTransfomer | 0.0100 | 0.0076 | 0.0071 | 0.0622 | 0.0052 | 0.0184±0.0110 | 0.0060 | 0.0346 | 0.0271 | 0.0283 | 0.1238 | 0.0200 | 0.0400±0.0172 | 0.0229 |
| Mamba | 0.0498 | 0.0033 | 0.0038 | 0.0626 | 0.0020 | 0.0243±0.0132 | 0.0020 | 0.0167 | 0.0127 | 0.0100 | 0.0980 | 0.0086 | 0.0247±0.0148 | 0.0111 |
| StageNet | 0.2029 | 0.2014 | 0.2028 | 0.2045 | 0.2004 | 0.2024±0.0007 | 0.1302 | 0.4055 | 0.4037 | 0.4051 | 0.4072 | 0.4025 | 0.4048±0.0008 | 0.2731 |
| AdaCare | 0.0099 | 0.0059 | 0.0085 | 0.0114 | 0.0046 | 0.0081±0.0013 | 0.0066 | 0.0115 | 0.0063 | 0.0090 | 0.0137 | 0.0045 | 0.0090±0.0016 | 0.0117 |

### IRWO

| Model | Fold 1 MSE | Fold 2 MSE | Fold 3 MSE | Fold 4 MSE | Fold 5 MSE | Mean MSE | Test MSE | Fold 1 MAE | Fold 2 MAE | Fold 3 MAE | Fold 4 MAE | Fold 5 MAE | Mean MAE | Test MAE |
|---|---|---|---|---|---|---|---|---|---|---|---|---|---|---|
| Decision Tree | 0.0144 | 0.0163 | 0.0167 | 0.0148 | 0.0159 | 0.0156±0.0004 | 0.0168 | 0.0144 | 0.0163 | 0.0167 | 0.0148 | 0.0159 | 0.0156±0.0004 | 0.0168 |
| Logistic Regression | 0.0458 | 0.0420 | 0.0442 | 0.0414 | 0.0394 | 0.0426±0.0011 | 0.0391 | 0.0458 | 0.0420 | 0.0442 | 0.0414 | 0.0394 | 0.0426±0.0011 | 0.0391 |
| Random Forest | 0.0077 | 0.0087 | 0.0100 | 0.0069 | 0.0076 | 0.0082±0.0005 | 0.0087 | 0.0159 | 0.0165 | 0.0186 | 0.0151 | 0.0158 | 0.0164±0.0006 | 0.0172 |
| SVM | 0.0930 | 0.0975 | 0.0949 | 0.0936 | 0.0984 | 0.0955±0.0011 | 0.0980 | 0.0930 | 0.0976 | 0.0951 | 0.0937 | 0.0984 | 0.0956±0.0011 | 0.0980 |
| MLP | 0.0180 | 0.0144 | 0.0166 | 0.0145 | 0.0149 | 0.0157±0.0007 | 0.0847 | 0.0718 | 0.0620 | 0.0659 | 0.0667 | 0.0655 | 0.0664±0.0016 | 0.1007 |
| XGBoost | 0.0088 | 0.0091 | 0.0109 | 0.0083 | 0.0085 | 0.0091±0.0005 | 0.0098 | 0.0271 | 0.0269 | 0.0286 | 0.0287 | 0.0275 | 0.0278±0.0004 | 0.0281 |
| LightGBM | 0.0078 | 0.0086 | 0.0069 | 0.0075 | 0.0075 | 0.0081±0.0008 | 0.0086 | 0.0188 | 0.0191 | 0.0211 | 0.0183 | 0.0190 | 0.0193±0.0005 | 0.0198 |
| AdaBoost | 0.0472 | 0.0484 | 0.0483 | 0.0503 | 0.0541 | 0.0497±0.0012 | 0.0471 | 0.1121 | 0.1151 | 0.1107 | 0.1179 | 0.1217 | 0.1155±0.0020 | 0.1122 |
| Transtab | 0.0807 | 0.0803 | 0.0766 | 0.0799 | 0.0816 | 0.0798±0.0009 | 0.0798 | 0.0807 | 0.0803 | 0.0766 | 0.0800 | 0.0813 | 0.0797±0.0008 | 0.0798 |
| iTransfomer | 0.0105 | 0.0052 | 0.0070 | 0.0124 | 0.0041 | 0.0078±0.0014 | 0.0060 | 0.0465 | 0.0293 | 0.0426 | 0.0651 | 0.0410 | 0.0384±0.0080 | 0.0300 |
| Mamba | 0.0079 | 0.0021 | 0.0027 | 0.0273 | 0.0269 | 0.0134±0.0057 | 0.0021 | 0.0174 | 0.0061 | 0.0144 | 0.0552 | 0.0572 | 0.0254±0.0100 | 0.0121 |
| StageNet | 0.0801 | 0.0802 | 0.0850 | 0.0859 | 0.0803 | 0.0823±0.0013 | 0.0819 | 0.1723 | 0.1720 | 0.1770 | 0.1776 | 0.1731 | 0.1744±0.0012 | 0.1732 |
| AdaCare | 0.0086 | 0.0096 | 0.0144 | 0.0151 | 0.0103 | 0.0116±0.0013 | 0.0079 | 0.0109 | 0.0119 | 0.0167 | 0.0174 | 0.0126 | 0.0139±0.0013 | 0.0085 |

### NSAID

| Model | Fold 1 MSE | Fold 2 MSE | Fold 3 MSE | Fold 4 MSE | Fold 5 MSE | Mean MSE | Test MSE | Fold 1 MAE | Fold 2 MAE | Fold 3 MAE | Fold 4 MAE | Fold 5 MAE | Mean MAE | Test MAE |
|---|---|---|---|---|---|---|---|---|---|---|---|---|---|---|
| Decision Tree | 0.1379 | 0.1425 | 0.1397 | 0.1370 | 0.1451 | 0.1404±0.0015 | 0.1493 | 0.1373 | 0.1419 | 0.1394 | 0.1358 | 0.1442 | 0.1397±0.0015 | 0.1489 |
| Logistic Regression | 0.0956 | 0.0941 | 0.0980 | 0.0975 | 0.0937 | 0.0958±0.0009 | 0.1011 | 0.0953 | 0.0938 | 0.0974 | 0.0966 | 0.0937 | 0.0954±0.0007 | 0.1004 |
| Random Forest | 0.0709 | 0.0684 | 0.0719 | 0.0722 | 0.0698 | 0.0706±0.0007 | 0.0745 | 0.1405 | 0.1370 | 0.1420 | 0.1427 | 0.1405 | 0.1405±0.0010 | 0.1461 |
| SVM | 0.1283 | 0.1297 | 0.1368 | 0.1307 | 0.1278 | 0.1307±0.0016 | 0.1324 | 0.1193 | 0.1186 | 0.1228 | 0.1201 | 0.1165 | 0.1195±0.0010 | 0.1235 |
| MLP | 0.0877 | 0.0835 | 0.0923 | 0.0858 | 0.0850 | 0.0869±0.0015 | 0.3490 | 0.1915 | 0.1900 | 0.2015 | 0.1932 | 0.1931 | 0.1939±0.0020 | 0.1985 |
| XGBoost | 0.0715 | 0.0722 | 0.0736 | 0.0735 | 0.0716 | 0.0725±0.0005 | 0.0766 | 0.1493 | 0.1484 | 0.1516 | 0.1521 | 0.1491 | 0.1501±0.0007 | 0.1532 |
| LightGBM | 0.0669 | 0.0662 | 0.0669 | 0.0676 | 0.0667 | 0.0674±0.0006 | 0.0714 | 0.1364 | 0.1347 | 0.1390 | 0.1386 | 0.1368 | 0.1371±0.0008 | 0.1415 |
| AdaBoost | 0.2356 | 0.2344 | 0.2234 | 0.2427 | 0.2342 | 0.2341±0.0031 | 0.2382 | 0.4850 | 0.4837 | 0.4714 | 0.4916 | 0.4836 | 0.4831±0.0033 | 0.4873 |
| Transtab | 0.2918 | 0.2944 | 0.2939 | 0.2929 | 0.2969 | 0.2940±0.0009 | 0.2940 | 0.2916 | 0.2930 | 0.2929 | 0.2925 | 0.2965 | 0.2928±0.0004 | 0.2933 |
| iTransfomer | 0.1054 | 0.0688 | 0.0816 | 0.1030 | 0.0746 | 0.0867±0.0074 | 0.0723 | 0.1537 | 0.1442 | 0.1759 | 0.2060 | 0.1466 | 0.1498±0.0183 | 0.1493 |
| Mamba | 0.0021 | 0.0635 | 0.1067 | 0.1400 | 0.0727 | 0.0770±0.0231 | 0.0766 | 0.0042 | 0.1244 | 0.1479 | 0.1980 | 0.1281 | 0.1132±0.0271 | 0.1432 |
| StageNet | 0.2044 | 0.2076 | 0.2114 | 0.2141 | 0.2114 | 0.2098±0.0017 | 0.1997 | 0.4102 | 0.4120 | 0.4131 | 0.4207 | 0.4184 | 0.4149±0.0020 | 0.4598 |
| AdaCare | 0.0927 | 0.0952 | 0.1018 | 0.1092 | 0.0976 | 0.0993±0.0029 | 0.0863 | 0.1186 | 0.1114 | 0.1134 | 0.1258 | 0.1289 | 0.1196±0.0034 | 0.1080 |

### A/A

| Model | Fold 1 MSE | Fold 2 MSE | Fold 3 MSE | Fold 4 MSE | Fold 5 MSE | Mean MSE | Test MSE | Fold 1 MAE | Fold 2 MAE | Fold 3 MAE | Fold 4 MAE | Fold 5 MAE | Mean MAE | Test MAE |
|---|---|---|---|---|---|---|---|---|---|---|---|---|---|---|
| Decision Tree | 0.0304 | 0.0323 | 0.0345 | 0.0338 | 0.0328 | 0.0328±0.0007 | 0.0352 | 0.0304 | 0.0317 | 0.0345 | 0.0335 | 0.0328 | 0.0326±0.0007 | 0.0352 |
| Logistic Regression | 0.0223 | 0.0216 | 0.0200 | 0.0235 | 0.0248 | 0.0224±0.0008 | 0.0245 | 0.0223 | 0.0216 | 0.0200 | 0.0232 | 0.0248 | 0.0224±0.0008 | 0.0245 |
| Random Forest | 0.0158 | 0.0162 | 0.0141 | 0.0168 | 0.0174 | 0.0161±0.0006 | 0.0180 | 0.0327 | 0.0317 | 0.0317 | 0.0331 | 0.0338 | 0.0326±0.0004 | 0.0363 |
| SVM | 0.0436 | 0.0436 | 0.0414 | 0.0432 | 0.0443 | 0.0432±0.0005 | 0.0442 | 0.0504 | 0.0437 | 0.0450 | 0.0427 | 0.0444 | 0.0452±0.0013 | 0.0442 |
| MLP | 0.0236 | 0.0196 | 0.0196 | 0.0222 | 0.0241 | 0.0218±0.0010 | 0.0541 | 0.0793 | 0.0682 | 0.0766 | 0.0771 | 0.0787 | 0.0760±0.0020 | 0.0691 |
| XGBoost | 0.0170 | 0.0166 | 0.0156 | 0.0187 | 0.0191 | 0.0174±0.0007 | 0.0199 | 0.0369 | 0.0349 | 0.0356 | 0.0371 | 0.0375 | 0.0364±0.0005 | 0.0392 |
| LightGBM | 0.0153 | 0.0153 | 0.0135 | 0.0160 | 0.0170 | 0.0154±0.0006 | 0.0173 | 0.0316 | 0.0313 | 0.0301 | 0.0317 | 0.0322 | 0.0314±0.0004 | 0.0338 |
| AdaBoost | 0.2207 | 0.0268 | 0.0607 | 0.0392 | 0.0391 | 0.0773±0.0363 | 0.2233 | 0.4649 | 0.0699 | 0.2013 | 0.1005 | 0.1144 | 0.1902±0.0721 | 0.4674 |
| Transtab | 0.0404 | 0.0398 | 0.0452 | 0.0465 | 0.0449 | 0.0434±0.0014 | 0.0434 | 0.0446 | 0.0398 | 0.0452 | 0.0463 | 0.0446 | 0.0430±0.0013 | 0.0433 |
| iTransfomer | 0.7698 | 0.0167 | 0.0214 | 0.0320 | 0.0176 | 0.1715±0.1496 | 0.0210 | 0.0735 | 0.0459 | 0.0533 | 0.0876 | 0.0420 | 0.0539±0.0097 | 0.0489 |
| Mamba | 0.0251 | 0.0155 | 0.0200 | 0.0268 | 0.0177 | 0.0210±0.0022 | 0.0271 | 0.0440 | 0.0500 | 0.0418 | 0.0531 | 0.0393 | 0.0426±0.0037 | 0.0426 |
| StageNet | 0.0421 | 0.0359 | 0.0412 | 0.0435 | 0.0368 | 0.0399±0.0015 | 0.0402 | 0.0822 | 0.0779 | 0.0836 | 0.0853 | 0.079 | 0.0816±0.0014 | 0.0478 |
| AdaCare | 0.0307 | 0.0219 | 0.0276 | 0.0283 | 0.0197 | 0.0257±0.0021 | 0.0345 | 0.0321 | 0.0231 | 0.0307 | 0.0313 | 0.0219 | 0.0279±0.0022 | 0.0439 |

### Others

| Model | Fold 1 MSE | Fold 2 MSE | Fold 3 MSE | Fold 4 MSE | Fold 5 MSE | Mean MSE | Test MSE | Fold 1 MAE | Fold 2 MAE | Fold 3 MAE | Fold 4 MAE | Fold 5 MAE | Mean MAE | Test MAE |
|---|---|---|---|---|---|---|---|---|---|---|---|---|---|---|
| Decision Tree | 0.0021 | 0.0038 | 0.0023 | 0.0028 | 0.0018 | 0.0026±0.0004 | 0.0050 | 0.0021 | 0.0032 | 0.0021 | 0.0025 | 0.0018 | 0.0023±0.0002 | 0.0050 |
| Logistic Regression | 0.0021 | 0.0028 | 0.0022 | 0.0021 | 0.0009 | 0.0020±0.0003 | 0.0007 | 0.0018 | 0.0025 | 0.0016 | 0.0018 | 0.0009 | 0.0017±0.0003 | 0.0007 |
| Random Forest | 0.0011 | 0.0017 | 0.0015 | 0.0015 | 0.0007 | 0.0013±0.0002 | 0.0012 | 0.0021 | 0.0027 | 0.0023 | 0.0024 | 0.0017 | 0.0022±0.0002 | 0.0032 |
| SVM | 0.0010 | 0.0016 | 0.0018 | 0.0018 | 0.0010 | 0.0014±0.0002 | 0.0006 | 0.0010 | 0.0016 | 0.0012 | 0.0015 | 0.0010 | 0.0013±0.0001 | 0.0006 |
| MLP | 0.0023 | 0.0020 | 0.0021 | 0.0020 | 0.0016 | 0.0020±0.0001 | 0.0064 | 0.0225 | 0.0204 | 0.0213 | 0.0186 | 0.0207 | 0.0207±0.0006 | 0.0175 |
| XGBoost | 0.0014 | 0.0025 | 0.0014 | 0.0018 | 0.0010 | 0.0016±0.0002 | 0.0017 | 0.0024 | 0.0035 | 0.0025 | 0.0027 | 0.0022 | 0.0027±0.0002 | 0.0039 |
| LightGBM | 0.0011 | 0.0015 | 0.0016 | 0.0015 | 0.0007 | 0.0013±0.0002 | 0.0009 | 0.0036 | 0.0036 | 0.0037 | 0.0036 | 0.0030 | 0.0035±0.0001 | 0.0039 |
| AdaBoost | 0.0718 | 0.0451 | 0.0045 | 0.0136 | 0.0787 | 0.0427±0.0149 | 0.0375 | 0.2213 | 0.1285 | 0.0133 | 0.0418 | 0.2395 | 0.1289±0.0457 | 0.1194 |
| Transtab | 0.0015 | 0.0007 | 0.0015 | 0.0015 | 0.0010 | 0.0012±0.0002 | 0.0024 | 0.0012 | 0.0007 | 0.0012 | 0.0012 | 0.0010 | 0.0011±0.0001 | 0.0011 |
| iTransfomer | 0.0030 | 0.0032 | 0.0029 | 0.0002 | 0.0005 | 0.0020±0.0007 | 0.0023 | 0.0047 | 0.0075 | 0.0074 | 0.0027 | 0.0070 | 0.0053±0.0010 | 0.0052 |
| Mamba | 0.2210 | 0.0021 | 0.0010 | 0.0010 | 0.0010 | 0.0452±0.0439 | 0.0023 | 0.0128 | 0.0050 | 0.0050 | 0.0050 | 0.0038 | 0.0055±0.0015 | 0.0060 |
| StageNet | 0.0004 | 0.0002 | 0.0009 | 0.0003 | 0.0011 | 0.0005±0.0002 | 0.0005 | 0.0004 | 0.0002 | 0.0009 | 0.0003 | 0.0011 | 0.0005±0.0002 | 0.0005 |
| AdaCare | 0.0003 | 0.0002 | 0.0007 | 0.0002 | 0.0008 | 0.0004±0.0002 | 0.0005 | 0.0003 | 0.0002 | 0.0007 | 0.0003 | 0.0014 | 0.0005±0.0003 | 0.0005 |

until its performance has been rigorously evaluated and the system's scope and representative-
ness in relation to real-world applications have been validated. Data usage must strictly adhere
to applicable regulations in China. Access to the PEACE dataset can be found at the following
address:[https://github.com/YTYTYD/PEACE].

### E.1 Dataset Documentation

**Main Data:**

1. All_Data.csv: a .CSV file containing all patients in the dataset, with patient ID.

2. All_data.json: a .JSON file describing all the data in the dataset.

**Dictionaries:**

1. D_ Numerical.csv: A .csv file containing the units of the numerical features.

2. D_ Multiclass.csv: A .csv file containing the meaning of multiclass features.

3. D_ Diagnosis.csv: A .csv file containing the meaning of diagnosis.

**Model Training:**

1. Train data: a .CSV file containing the training set of patients.

2. Test data: a .CSV file containing the test set of patients.

### E.2 Responsibility Statement

The corresponding author(s) acknowledge and accept full responsibility for any potential infringement
of rights associated with this dataset.

### E.3 Ethical Considerations

All data are de-identified to the greatest extent possible and stored in a database controlled internally
by Xiangya Hospital. This work has been approved by the Xiangya Hospital Institutional Review
Board (Ethics Approval No.: 202109422). The data are available for future research by other Xiangya
Hospital researchers. Access for external researchers will be provided under restricted conditions,
with permissions ultimately reviewed by the Xiangya Hospital.

## F Samples and Case Studies

**Sample 1:**
As shown in Table 16, the patient in Sample 1 was diagnosed with a malignant tumor of the right
kidney with multiple metastases. The patient denies any history of allergies, smoking, or alcohol
consumption. Chemotherapy was chosen as the treatment method for the tumor. After evaluation,
no cardiovascular or gastrointestinal risks were identified. The results of the complete blood count,
liver function, and kidney function tests were all within normal ranges. The type of pain experienced
is somatic, with a Numerical Rating Scale (NRS) score of 8 at its most severe, 6 at its least severe,
an average of 8, and currently 6. This indicates severe pain that significantly affects the patient's
daily life and emotions. The pain occurs three or more times per day. Breakthrough pain is of the
end-of-dose type, occurring three or more times per day. The tumor symptoms are severe. The
patient has been using sustained-release strong opioids for three days, with a compliance score of
5.75, and has not tolerated opioids well. Pain control lasts for six hours post-medication, with side
effects of constipation, nausea, and vomiting, which have been managed with additional medications.
The patient's pain control is poor, possibly due to inappropriate medication selection. The doctor
and pharmacist recommended continuing the use of sustained-release strong opioids and adding
NSAIDs, along with medications for constipation and nausea. The patient fully complied with and

followed the advice. One week later, during follow-up, the pain was mildly relieved and evaluated as moderate pain. It was recommended to increase the dose of sustained-release strong opioids, continue using NSAIDs, and medications for adverse effects. After adjusting the dose, the pain was partially relieved, but breakthrough pain persisted. It was recommended to use sustained-release strong opioids, immediate-release strong opioids, and NSAIDs. Following this adjustment, the patient's pain was completely relieved, and it was recommended to continue the treatment as per the original plan.

**Sample 2:**

As shown in Table 17, the patient in Sample 2 was diagnosed with a malignant tumor of the jejunum. The patient denies any history of allergies, smoking, or alcohol consumption. The treatment for the tumor involved surgery. After evaluation, there were no cardiovascular or gastrointestinal risks. The results of the complete blood count, liver function, and kidney function tests were all normal. The type of pain experienced is visceral pain, with an NRS (Numerical Rating Scale) of 6 at its most severe, 3 at its least severe, an average of 5, and currently 2. The pain affects daily life and emotions. The frequency of pain is less than three times per day, with activity-induced breakthrough pain occurring less than three times per day. The tumor symptoms are mild. The patient has been using immediate-release weak opioids for 10 days, with a compliance score of 3.25. Nausea and vomiting were observed after medication administration. Poor pain control might be due to an insufficient dose. The pharmacist and doctor recommended continuing the use of immediate-release weak opioids and increasing the dose, along with antiemetic medication. After administration, the pain was partially relieved. Five days later, the patient's NRS was 7 at its most severe, 4 at its least severe, with an average of 6, and currently 6. No breakthrough pain was reported. The patient had been using immediate-release weak opioids for 15 days, with a compliance score of 7. The analgesic effect was poor, possibly due to inappropriate medication selection. After discussion with the pharmacist, the doctor adjusted the medication to sustained-release strong opioids. The patient fully complied and followed the advice. One week later, during follow-up, the pain was partially relieved after taking sustained-release strong opioids.

**Sample 3:**

As shown in Table 18, the patient in Sample 3 was diagnosed with a malignant tumor of the ascending colon. The patient denies any history of allergies or smoking but has a history of alcohol consumption. After evaluation, there were no cardiovascular or gastrointestinal risks. The results of the complete blood count, liver function, and kidney function tests were all normal. The type of pain is mixed, with an NRS (Numerical Rating Scale) of 10 at its most severe, 2 at its least severe, an average of 6, and currently 8. The pain affects daily life and emotions. The pain frequency is less than three times per day, with breakthrough pain of the end-of-dose type occurring three or more times per day. The tumor symptoms are severe. Currently, the patient is not using any analgesic medication. The pharmacist and doctor recommended immediate-release weak opioids, which partially relieved the pain after administration. One week later, the patient's NRS was 4 at its most severe, 2 at its least severe, with an average of 3, and currently 2. The pain has a slight impact on daily life and emotions, with no breakthrough pain. The patient has been using immediate-release weak opioids for 7 days, with a compliance score of 6.5. After medication, pain control lasts for 5 hours, with no adverse reactions observed. The analgesic effect is poor, possibly due to inappropriate medication selection. After discussion with the pharmacist, the doctor adjusted the medication to sustained-release strong opioids. The patient fully complied and followed the advice. One week later, during follow-up, the patient's pain was completely relieved after taking sustained-release strong opioids.

**Sample 4:**

As shown in Table 19, the patient in Sample 4 was diagnosed with a malignant neck tumor. The patient denies any history of smoking, allergies, or alcohol consumption. Upon evaluation, there were no cardiovascular or gastrointestinal risks identified. Results from the complete blood count, liver function, and kidney function tests were all within normal ranges. The patient's pain is characterized as somatic, with a Numerical Rating Scale (NRS) score of 10 at its most severe, 6 at its least severe, an average of 7, and currently 5. The pain significantly impacts daily life and emotional well-being and is persistent. The patient experiences breakthrough pain less than three times per day, primarily activity-

## Table 16: Sample 1

**Patient Basic Information**

| ID | Gender | Age | Height | Weight |
|---|---|---|---|---|
| SJ-289031 | 1 | 59 | 170 | 75 |
| SJ-289031 | 1 | 59 | 170 | 75 |
| SJ-289031 | 1 | 59 | 170 | 75 |
| SJ-289031 | 1 | 59 | 170 | 75 |

| BMI | Body Surface Area (BSA) | Medical Record Date | Length of Hospital Stay | Number of Hospital Admissions |
|---|---|---|---|---|
| 25.95 | 1.8441 | 2050/2/10 | 1 | 1 |
| 25.95 | 1.8441 | 2050/2/12 | 3 | 2 |
| 25.95 | 1.8441 | 2050/2/19 | 10 | 3 |
| 25.95 | 1.8441 | 2050/2/26 | 17 | 4 |

| Diagnosis | Smoking History | Drinking History | Allergy History | Tumour Treatment Methods |
|---|---|---|---|---|
| 112 | 0 | 0 | 0 | 2 |
| 112 | 0 | 0 | 0 | 2 |
| 112 | 0 | 0 | 0 | 2 |
| 112 | 0 | 0 | 0 | 2 |

| Cardiovascular Risk | Gastrointestinal Risk | PS Score | White Blood Cell Count | Red Blood Cell Count |
|---|---|---|---|---|
| 0 | 0 | 3 | 7.5 | 5.3 |
| 0 | 0 | 2 | 4.2 | 4.62 |
| 0 | 0 | 2 | 5.6 | 3.84 |
| 0 | 0 | 2 | 4.7 | 5.17 |

| Hemoglobin | Platelet Count | Hematocrit | Neutrophil Count | Lymphocyte Count |
|---|---|---|---|---|
| 162 | 130 | 48.2 | 4.4 | 1.8 |
| 140 | 184 | 42.1 | 2.3 | 1.5 |
| 120 | 146 | 34.5 | 4.2 | 1 |
| 150 | 131 | 45.8 | 2.2 | 1.9 |

| Eosinophil Count | Basophil Count | Monocyte Percentage | Neutrophil Percentage | Lymphocyte Percentage |
|---|---|---|---|---|
| 0.43 | 0.06 | 10.5 | 58.9 | 24 |
| 0.1 | 0 | 8.2 | 54.3 | 35 |
| 0 | 0 | 6.8 | 75.7 | 17.4 |
| 0.08 | 0.02 | 11.8 | 46.4 | 39.7 |

| Basophil Percentage | Eosinophil Percentage | Mean Corpuscular Volume | Mean Corpuscular Hemoglobin | Mean Corpuscular Hemoglobin Concentration |
|---|---|---|---|---|
| 0.9 | 5.8 | 90.8 | 30.6 | 336.7 |
| 0.7 | 1.8 | 91 | 30.3 | 332.5 |
| 0.1 | 0 | 89.7 | 31.3 | 348.6 |
| 0.4 | 1.7 | 88.6 | 29 | 328 |

| Red Cell Distribution Width | Plateletcrit | Mean Platelet Volume | Total Protein | Albumin |
|---|---|---|---|---|
| 14.5 | 0.13 | 10.1 | 67.6 | 38.8 |
| 13.2 | 0.15 | 8.36 | 63.5 | 40.5 |
| 14.1 | 0.04 | 8.65 | 61.4 | 41.4 |
| 13.9 | 0.15 | 11.4 | 54.4 | 36.9 |

| Globulin | Albumin/Globulin Ratio | Total Bilirubin | Direct Bilirubin | Total Bile Acids |
|---|---|---|---|---|
| 28.8 | 1.3 | 14.5 | 6.8 | 5.5 |
| 23 | 1.8 | 7.3 | 3.9 | 3.4 |
| 20 | 2.1 | 4.8 | 1.3 | 3.2 |
| 17.5 | 2.1 | 17.7 | 6.3 | 8.1 |

| Alanine Aminotransferase | Aspartate Aminotransferase | Urea | Creatinine | Uric Acid |
|---|---|---|---|---|
| 17.4 | 17.5 | 5.38 | 88 | 421.8 |
| 27.8 | 17.5 | 5.63 | 78 | 381.5 |
| 12.6 | 11.4 | 4.26 | 68.1 | 291.3 |
|  |  | 7.5 | 58.6 | 345.4 |

**Comprehensive Pain Assessment**

| Pain Type | Worst Pain | Mildest Pain | Average Pain | Current Pain |
|---|---|---|---|---|
| 2 | 8 | 6 | 8 | 6 |
| 2 | 6 | 4 | 6 | 3 |
| 2 | 6 | 2 | 2 | 1 |
| 2 | 1 | 0 | 1 | 0 |

| Impact of Pain on Daily Life | Impact of Pain on Mood | Impact of Pain on Walking Ability | Impact of Pain on Daily Work | Impact of Pain on Relationships with Others |
|---|---|---|---|---|
| 7 | 7 | 7 | 9 | 1 |
| 3 | 4 | 7 | 6 | 0 |
| 1 | 0 | 1 | 3 | 0 |
| 1 | 2 | 0 | 4 | 0 |

| Impact of Pain on Sleep | Impact of Pain on Interest in Life | Pain Frequency | Type of Breakthrough Pain | Frequency of Breakthrough Pain |
|---|---|---|---|---|
| 10 | 6 | 2 | 2 | 2 |
| 5 | 0 | 2 | 2 | 1 |
| 1 | 0 | 1 | 2 | 2 |
| 1 | 0 | 0 | 0 | 0 |

**Previous Analgesic Treatment**

| Prev_ERSO | Prev_IRSO | Prev_ERWO | Prev_IRWO | Prev_NSAID |
|---|---|---|---|---|
| 1 | 0 | 0 | 0 | 0 |
| 1 | 0 | 0 | 0 | 1 |
| 1 | 0 | 0 | 0 | 1 |
| 1 | 1 | 0 | 0 | 1 |

| Prev_A/A | Prev_Others | Opiate Tolerance | Days of Medication Use | M1 |
|---|---|---|---|---|
| 0 | 0 | 0 | 3 | 1 |
| 0 | 0 | 0 | 5 | 1 |
| 0 | 0 | 0 | 12 | 1 |
| 0 | 0 | 0 | 19 | 1 |

| M2 | M3 | M4 | M5 | M6 |
|---|---|---|---|---|
| 1 | 1 | 0 | 1 | 1 |
| 1 | 1 | 1 | 1 | 1 |
| 1 | 1 | 1 | 1 | 1 |
| 1 | 1 | 1 | 1 | 1 |

| M7 | M8 | MMAS-8 Total Score | Duration of Analgesic Control | Constipation |
|---|---|---|---|---|
| 0 | 0.75 | 5.75 | 6 | 1 |
| 1 | 1 | 8 | 8 | 1 |
| 1 | 1 | 8 | 8 | 1 |
| 1 | 1 | 8 | 12 | 1 |

| Nausea/Vomiting | Other Adverse Reactions | Medication for Adverse Reactions |
|---|---|---|
| 1 | 0 | 1 |
| 1 | 0 | 1 |
| 0 | 0 | 1 |
| 0 | 0 | 1 |

**Cancer Pain Medication Decision**

| ERSO_Recom | IRSO_Recom | ERWO_Recom | IRWO_Recom | NSAIDs_Recom |
|---|---|---|---|---|
| 1 | 0 | 0 | 0 | 0 |
| 1 | 0 | 0 | 0 | 1 |
| 1 | 0 | 0 | 0 | 1 |
| 1 | 0 | 0 | 0 | 1 |

| A/A_Recom | Others_Recom | Constipation Medication Recommended | Nausea/Vomiting Medication Recommended |
|---|---|---|---|
| 0 | 0 | 2 | 1 |
| 0 | 0 | 2 | 1 |
| 0 | 0 | 2 | 0 |
| 0 | 0 | 2 | 0 |

**Evaluation of Previous Analgesic Treatment**

| Drug-Related Problems | Causes | Interventions | Acceptance of Interventions | Status of DRPs |
|---|---|---|---|---|
| 2 | 1 | 15 | 1 | 3 |
| 2 | 9 | 11 | 1 | 3 |
| 2 | 9 | 10 | 1 | 3 |
| 0 | 0 | 0 | 0 | 1 |

**Follow-up**

| Pain Relief Status |
|---|
| 3 |
| 2 |
| 2 |
| 1 |

## Table 17: Sample 2

**Patient Basic Information**

| ID | Gender | Age | Height | Weight |
|---|---|---|---|---|
| SJ-514441 | 0 | 53 | 152 | 36 |
| SJ-514441 | 0 | 53 | 152 | 36 |

| BMI | Body Surface Area (BSA) | Medical Record Date | Length of Hospital Stay | Number of Hospital Admissions |
|---|---|---|---|---|
| 1.2351 | 2052/2/3 | 2 | 2 |  |
| 1.2351 | 2052/4/11 | 2 | 5 |  |

| Diagnosis | Smoking History | Drinking History | Allergy History | Tumour Treatment Methods |
|---|---|---|---|---|
| 54 | 0 | 0 | 0 | 1 |
| 54 | 0 | 0 | 0 | 1 |

| Cardiovascular Risk | Gastrointestinal Risk | PS Score | White Blood Cell Count | Red Blood Cell Count |
|---|---|---|---|---|
| 0 | 0 | 1 | 10.8 | 5.43 |
| 0 | 0 | 2 | 7.1 | 4.98 |

| Hemoglobin | Platelet Count | Hematocrit | Neutrophil Count | Lymphocyte Count |
|---|---|---|---|---|
| 133 | 175 | 36.2 | 5.4 | 1.3 |
| 141 | 128 | 43.5 | 5.2 | 1.3 |

| Eosinophil Count | Basophil Count | Monocyte Percentage | Neutrophil Percentage | Lymphocyte Percentage |
|---|---|---|---|---|
| 0 | 0 | 0.5 | 89 | 10.4 |
| 0 | 0.1 | 7.6 | 73.3 | 18.1 |

| Basophil Percentage | Eosinophil Percentage | Mean Corpuscular Volume | Mean Corpuscular Hemoglobin | Mean Corpuscular Hemoglobin Concentration |
|---|---|---|---|---|
| 0.1 | 0 | 66.7 | 20.7 | 311 |
| 0.7 | 0.3 | 67.2 | 20.9 | 311.6 |

| Red Cell Distribution Width | Plateletcrit | Mean Platelet Volume | Total Protein | Albumin |
|---|---|---|---|---|
| 16.8 | 0.18 | 10.4 |  |  |
| 16.2 | 0.12 | 9.63 | 64.7 | 39.9 |

| Globulin | Albumin/Globulin Ratio | Total Bilirubin | Direct Bilirubin | Total Bile Acids |
|---|---|---|---|---|
| 24.8 | 1.6 | 15.3 | 4.4 | 3.1 |

| Alanine Aminotransferase | Aspartate Aminotransferase | Urea | Creatinine | Uric Acid |
|---|---|---|---|---|
| 26.7 | 27.5 | 5.5 | 67 | 379.8 |

**Comprehensive Pain Assessment**

| Pain Type | Worst Pain | Mildest Pain | Average Pain | Current Pain |
|---|---|---|---|---|
| 1 | 6 | 3 | 5 | 6 |
| 1 | 7 | 4 | 6 | 6 |

| Impact of Pain on Daily Life | Impact of Pain on Mood | Impact of Pain on Walking Ability | Impact of Pain on Daily Work | Impact of Pain on Relationships with Others |
|---|---|---|---|---|
| 3 | 3 | 3 | 4 | 5 |
| 5 | 5 | 3 | 3 | 5 |

| Impact of Pain on Sleep | Impact of Pain on Interest in Life | Pain Frequency | Type of Breakthrough Pain | Frequency of Breakthrough Pain |
|---|---|---|---|---|
| 5 | 3 | 0 | 1 | 1 |
| 5 | 4 | 0 | 0 | 0 |

**Previous Analgesic Treatment**

| Prev_ERSO | Prev_IRSO | Prev_ERWO | Prev_IRWO | Prev_NSAID |
|---|---|---|---|---|
| 0 | 0 | 0 | 1 | 0 |
| 0 | 0 | 0 | 1 | 0 |

| Prev_A/A | Prev_Others | Opiate Tolerance | Days of Medication Use | M1 |
|---|---|---|---|---|
| 0 | 0 | 0 | 10 | 0 |
| 0 | 1 | 0 | 15 | 0 |

| M2 | M3 | M4 | M5 | M6 |
|---|---|---|---|---|
| 1 | 1 | 0 | 0 | 0 |
| 1 | 1 | 1 | 1 | 1 |

| M7 | M8 | MMAS-8 Total Score | Duration of Analgesic Control | Constipation |
|---|---|---|---|---|
| 1 | 0.25 | 3.25 | 7 | 0 |
| 1 | 1 | 7 | 6 | 0 |

| Nausea/Vomiting | Other Adverse Reactions | Medication for Adverse Reactions | | |
|---|---|---|---|---|
| 1 | 0 | 0 |  |  |
| 1 | 0 | 0 |  |  |

**Cancer Pain Medication Decision**

| ERSO_Recom | IRSO_Recom | ERWO_Recom | IRWO_Recom | NSAIDs_Recom |
|---|---|---|---|---|
| 0 | 0 | 0 | 1 | 0 |
| 1 | 0 | 0 | 0 | 0 |

| A/A_Recom | Others_Recom | Constipation Medication Recommended | Nausea/Vomiting Medication Recommended | |
|---|---|---|---|---|
| 0 | 0 | 0 |  | 1 |
| 0 | 0 | 0 |  |  |

**Evaluation of Previous Analgesic Treatment**

| Drug-Related Problems | Causes | Interventions | Acceptance of Interventions | Status of DRPs |
|---|---|---|---|---|
| 2 | 9 | 11 | 1 | 2 |
| 2 | 1 | 10 | 2 | 3 |

**Follow-up**

| Pain Relief Status |
|---|
| 2 |
| 2 |

## Table 18: Sample 3

**Patient Basic Information**

| ID | Gender | Age | Height | Weight |
|---|---|---|---|---|
| SJ-921252 | 1 | 81 | 162 | 60 |
| SJ-921252 | 1 | 80 | 162 | 60 |

| BMI | Body Surface Area (BSA) | Medical Record Date | Length of Hospital Stay | Number of Hospital Admissions |
|---|---|---|---|---|
| | 1.2351 | 2074/10/20 | 11 | 2 |
| | 1.2351 | 2073/8/13 | 6 | 1 |

| Diagnosis | Smoking History | Drinking History | Allergy History | Tumour Treatment Methods |
|---|---|---|---|---|
| 744 | 0 | 1 | 0 | |
| 744 | 0 | 1 | 0 | |

| Cardiovascular Risk | Gastrointestinal Risk | PS Score | White Blood Cell Count | Red Blood Cell Count |
|---|---|---|---|---|
| 0 | 0 | 3 | 4.9 | 4.11 |
| 0 | 0 | 0 | 5.5 | 4.08 |

| Hemoglobin | Platelet Count | Hematocrit | Neutrophil Count | Lymphocyte Count |
|---|---|---|---|---|
| 145 | 145 | 39 | 7 | |
| 137 | 177 | 41.3 | 4.1 | 0.9 |

| Eosinophil Count | Basophil Count | Monocyte Percentage | Neutrophil Percentage | Lymphocyte Percentage |
|---|---|---|---|---|
| 0 | 0 | 8.3 | 73.5 | 17.1 |

| Basophil Percentage | Eosinophil Percentage | Mean Corpuscular Volume | Mean Corpuscular Hemoglobin | Mean Corpuscular Hemoglobin Concentration |
|---|---|---|---|---|
| 0.3 | 0.8 | 101.2 | 33.6 | 332 |

| Red Cell Distribution Width | Plateletcrit | Mean Platelet Volume | Total Protein | Albumin |
|---|---|---|---|---|
| 14.8 | 0.08 | 10.32 | 70 | 40.9 |

| Globulin | Albumin/Globulin Ratio | Total Bilirubin | Direct Bilirubin | Total Bile Acids |
|---|---|---|---|---|
| 29.1 | 1.4 | 18.3 | 5.2 | 9.7 |

| Alanine Aminotransferase | Aspartate Aminotransferase | Urea | Creatinine | Uric Acid |
|---|---|---|---|---|
| | | 5.76 | 58 | 232.1 |

**Comprehensive Pain Assessment**

| Pain Type | Worst Pain | Mildest Pain | Average Pain | Current Pain |
|---|---|---|---|---|
| 4 | 10 | 2 | 6 | 8 |
| 1 | 4 | 2 | 3 | 2 |

| Impact of Pain on Daily Life | Impact of Pain on Mood | Impact of Pain on Walking Ability | Impact of Pain on Daily Work | Impact of Pain on Relationships with Others |
|---|---|---|---|---|
| 4 | 5 | 3 | 4 | 4 |
| 1 | 0 | 0 | 0 | 2 |

| Impact of Pain on Sleep | Impact of Pain on Interest in Life | Pain Frequency | Type of Breakthrough Pain | Frequency of Breakthrough Pain |
|---|---|---|---|---|
| 5 | 4 | 1 | 2 | 2 |
| 2 | 1 | 0 | 0 | 0 |

**Previous Analgesic Treatment**

| Prev_ERSO | Prev_IRSO | Prev_ERWO | Prev_IRWO | Prev_NSAID |
|---|---|---|---|---|
| 0 | 0 | 0 | 0 | 0 |
| 0 | 0 | 0 | 1 | 0 |

| Prev_A/A | Prev_Others | Opiate Tolerance | Days of Medication Use | M1 |
|---|---|---|---|---|
| 0 | 0 | 0 | | |
| 0 | 0 | 0 | 7 | 1 |

| M2 | M3 | M4 | M5 | M6 |
|---|---|---|---|---|
| 1 | 1 | 1 | 1 | 1 |

| M7 | M8 | MMAS-8 Total Score | Duration of Analgesic Control | Constipation |
|---|---|---|---|---|
| | | | | 0 |
| 0 | 0.5 | 6.5 | 5 | 0 |

| Nausea/Vomiting | Other Adverse Reactions | Medication for Adverse Reactions | | |
|---|---|---|---|---|
| 0 | 0 | 0 | | |
| 0 | 0 | 0 | | |

**Cancer Pain Medication Decision**

| ERSO_Recom | IRSO_Recom | ERWO_Recom | IRWO_Recom | NSAIDs_Recom |
|---|---|---|---|---|
| 0 | 0 | 0 | 1 | 0 |
| 1 | 0 | 0 | 0 | 0 |

| A/A_Recom | Others_Recom | Constipation Medication Recommended | Nausea/Vomiting Medication Recommended | |
|---|---|---|---|---|
| 0 | 0 | | | |
| 0 | 0 | | | |

**Evaluation of Previous Analgesic Treatment**

| Drug-Related Problems | Causes | Interventions | Acceptance of Interventions | Status of DRPs |
|---|---|---|---|---|
| 2 | 1 | 10 | 1 | 2 |

**Follow-up**

| Pain Relief Status | | | | |
|---|---|---|---|---|
| 2 | | | | |
| 1 | | | | |

induced. The tumor symptoms are severe. Currently, the patient is on non-steroidal anti-inflammatory drugs (NSAIDs) and has been on this medication for 5 days, achieving a compliance score of 7.75. Pain relief lasts less than 1 hour after taking analgesics, with no adverse reactions reported. The analgesic effect is poor, possibly due to inappropriate medication selection. Following a discussion with the pharmacist, the physician adjusted the medication regimen to include sustained-release strong opioids combined with NSAIDs. The patient fully adhered to and followed the prescribed advice. One week later, during a follow-up visit, the patient's pain was completely relieved after medication.

Table 19: Sample 4

| **Patient Basic Information** | | | | |
|---|---|---|---|---|
| ID | Gender | Age | Height | Weight |
| SJ-854841 | 0 | 56 | 165 | 65 |
| BMI | Body Surface Area (BSA) | Medical Record Date | Length of Hospital Stay | Number of Hospital Admissions |
| | | 2089/5/31 | 13 | 1 |
| Diagnosis | Smoking History | Drinking History | Allergy History | Tumour Treatment Methods |
| 27 | 0 | 0 | 0 | |
| Cardiovascular Risk | Gastrointestinal Risk | PS Score | White Blood Cell Count | Red Blood Cell Count |
| 0 | 0 | 3 | 6.5 | 4.42 |
| Hemoglobin | Platelet Count | Hematocrit | Neutrophil Count | Lymphocyte Count |
| 138 | 250 | 41 | 4.7 | 1.1 |
| Eosinophil Count | Basophil Count | Monocyte Percentage | Neutrophil Percentage | Lymphocyte Percentage |
| 0.12 | 0.03 | 7 | 73.1 | 17.7 |
| Basophil Percentage | Eosinophil Percentage | Mean Corpuscular Volume | Mean Corpuscular Hemoglobin | Mean Corpuscular Hemoglobin Concentration |
| 0.4 | 1.8 | 92.8 | 31.2 | 336.6 |
| Red Cell Distribution Width | Plateletcrit | Mean Platelet Volume | Total Protein | Albumin |
| 14 | 0.22 | 8.64 | 67.8 | 42.7 |
| Globulin | Albumin/Globulin Ratio | Total Bilirubin | Direct Bilirubin | Total Bile Acids |
| 25.1 | 1.7 | 12 | 6.2 | 4.8 |
| Alanine Aminotransferase | Aspartate Aminotransferase | Urea | Creatinine | Uric Acid |
| 15.4 | 17.7 | 4.67 | 57 | 257 |
| **Comprehensive Pain Assessment** | | | | |
| Pain Type | Worst Pain | Mildest Pain | Average Pain | Current Pain |
| 2 | 10 | 6 | 7 | 5 |
| Impact of Pain on Daily Life | Impact of Pain on Mood | Impact of Pain on Walking Ability | Impact of Pain on Daily Work | Impact of Pain on Relationships with Others |
| 9 | 6 | 10 | 9 | 10 |
| Impact of Pain on Sleep | Impact of Pain on Interest in Life | Pain Frequency | Type of Breakthrough Pain | Frequency of Breakthrough Pain |
| 10 | 10 | 3 | 1 | 1 |
| **Previous Analgesic Treatment** | | | | |
| Prev_ERSO | Prev_IRSO | Prev_ERWO | Prev_IRWO | Prev_NSAID |
| 0 | 0 | 0 | 0 | 1 |
| Prev_A/A | Prev_Others | Opiate Tolerance | Days of Medication Use | M1 |
| 0 | 0 | 0 | 5 | 1 |
| M2 | M3 | M4 | M5 | M6 |
| 1 | 1 | 1 | 1 | 1 |
| M7 | M8 | MMAS-8 Total Score | Duration of Analgesic Control | Constipation |
| 1 | 0.75 | 7.75 | 1 | 0 |
| Nausea/Vomiting ons | Other Adverse Reacti Medication for Adverse Reactions | | | |
| 0 | 0 | 0 | | |
| **Cancer Pain Medication Decision** | | | | |
| ERSO_Recom | IRSO_Recom | ERWO_Recom | IRWO_Recom | NSAIDs_Recom |
| 1 | 0 | 0 | 0 | 1 |
| A/A_Recom | Others_Recom | Constipation Medication Recommended | Nausea/Vomiting Medication Recommended | |
| 0 | 0 | | | |
| **Evaluation of Previous Analgesic Treatment** | | | | |
| Drug-Related Problems | Causes | Interventions | Acceptance of Interventions | Status of DRPs |
| 2 | 1 | 10 | 1 | 3 |
| **Follow-up** | | | | |
| Pain Relief Status | | | | |
| 1 | | | | |

**Sample 5:**
As shown in Table 20, the patient in Sample 5 was diagnosed with adenocarcinoma of the upper left lung. The patient denies any history of allergies or alcohol consumption but has a history of smoking. Cardiovascular and gastrointestinal evaluations revealed no risks. Complete blood count, liver function, and kidney function tests were all normal. The patient reports experiencing visceral pain, with a Numerical Rating Scale (NRS) score of 10 at its most severe, 5 at its least severe, an average of 7, and a current score of 5. This pain significantly affects daily life and emotions and is persistent. The patient experiences breakthrough pain less than three times per day, classified as end-of-dose pain. The tumor symptoms are tolerable. Currently, the patient is using immediate-release weak opioids and has been on this medication for 31 days, with a compliance score of 7. Pain control

lasts for 5 hours after taking the analgesics, with no adverse reactions observed. The analgesic effect
is poor, possibly due to inappropriate medication selection. After consultation with the pharmacist,
the doctor adjusted the medication to sustained-release strong opioids. The patient fully complied
with the new regimen. One week later, during follow-up, the patient reported complete pain relief
after taking the sustained-release strong opioids.

Table 20: Sample 5

| **Patient Basic Information** | | | | |
| --- | --- | --- | --- | --- |
| ID | Gender | Age | Height | Weight |
| SJ-996524 | 1 | 40 | 172 | 49 |
| BMI | Body Surface Area (BSA) | Medical Record Date | Length of Hospital Stay | Number of Hospital Admissions |
| | | 2100/6/17 | 5 | 1 |
| Diagnosis | Smoking History | Drinking History | Allergy History | Tumour Treatment Methods |
| 118 | 1 | 0 | 0 | |
| Cardiovascular Risk | Gastrointestinal Risk | PS Score | White Blood Cell Count | Red Blood Cell Count |
| 0 | | 2 | 9.2 | 4.3 |
| Hemoglobin | Platelet Count | Hematocrit | Neutrophil Count | Lymphocyte Count |
| 127 | 391 | 36.4 | 6.7 | 0.5 |
| Eosinophil Count | Basophil Count | Monocyte Percentage | Neutrophil Percentage | Lymphocyte Percentage |
| 0.91 | 0.07 | 9.5 | 85.7 | 1.7 |
| Basophil Percentage | Eosinophil Percentage | Mean Corpuscular Volume | Mean Corpuscular Hemoglobin | Mean Corpuscular Hemoglobin Concentration |
| 0.2 | 2.9 | 84.7 | 27.2 | 321 |
| Red Cell Distribution Width | Plateletcrit | Mean Platelet Volume | Total Protein | Albumin |
| 14.4 | 0.32 | 8.1 | 58.2 | 31.4 |
| Globulin | Albumin/Globulin Ratio | Total Bilirubin | Direct Bilirubin | Total Bile Acids |
| 26.8 | 1.2 | 7.8 | 2.5 | 3.7 |
| Alanine Aminotransferase | Aspartate Aminotransferase | Urea | Creatinine | Uric Acid |
| | | 2.72 | 44 | 125.9 |
| **Comprehensive Pain Assessment** | | | | |
| Pain Type | Worst Pain | Mildest Pain | Average Pain | Current Pain |
| 1 | 10 | 5 | 7 | 5 |
| Impact of Pain on Daily Life | Impact of Pain on Mood | Impact of Pain on Walking Ability | Impact of Pain on Daily Work | Impact of Pain on Relationships with Others |
| 10 | 7 | 6 | 10 | 9 |
| Impact of Pain on Sleep | Impact of Pain on Interest in Life | Pain Frequency | Type of Breakthrough Pain | Frequency of Breakthrough Pain |
| 7 | 6 | 0 | 2 | 1 |
| **Previous Analgesic Treatment** | | | | |
| Prev_ERSO | Prev_IRSO | Prev_ERWO | Prev_IRWO | Prev_NSAID |
| 0 | 0 | 1 | 0 | 0 |
| Prev_A/A | Prev_Others | Opiate Tolerance | Days of Medication Use | M1 |
| 0 | 0 | 0 | 3 | 1 |
| M2 | M3 | M4 | M5 | M6 |
| 1 | 1 | 1 | 1 | 1 |
| M7 | M8 | MMAS-8 Total Score | Duration of Analgesic Control | Constipation |
| 1 | 0 | 7 | 5 | 0 |
| Nausea/Vomiting | Other Adverse Reactions | Medication for Adverse Reactions | | |
| 0 | 0 | 0 | | |
| **Cancer Pain Medication Decision** | | | | |
| ERSO_Recom | IRSO_Recom | ERWO_Recom | IRWO_Recom | NSAIDs_Recom |
| 1 | 0 | 0 | 0 | 1 |
| A/A_Recom | Others_Recom | Constipation Medication Recommended | Nausea/Vomiting Medication Recommended | |
| 0 | 0 | | | |
| **Evaluation of Previous Analgesic Treatment** | | | | |
| Drug-Related Problems | Causes | Interventions | Acceptance of Interventions | Status of DRPs |
| 2 | 1 | 10 | 1 | 3 |
| **Follow-up** | | | | |
| Pain Relief Status | | | | |
| 1 | | | | |

