# OpenReview forum: "PEACE: A Dataset of Pharmaceutical Care for Cancer Pain Analgesia Evaluation and Medication Decision"
_NeurIPS.cc/2024/Datasets_and_Benchmarks_Track — NeurIPS 2024 Track Datasets and Benchmarks Poster_

### Official Review · Reviewer_HMu4 · 2024-07-22
**Review of PEACE: A Dataset of Pharmaceutical Care for Cancer Pain Analgesia Evaluation and Medication Decision**

**Rating:** 6
**Confidence:** 2

**Review:**

The PEACE dataset is a valuable addition to cancer pain management research, offering extensive and diverse patient records, including long-term follow-ups and multidisciplinary treatment assessments. The dataset's comprehensive nature and effective de-identification efforts are commendable, though further clarification on the inclusion of medical professionals' rationales behind medication plans would enhance clarity. The inability to access the dataset due to ethical constraints limits the assessment of data quality. Currently, I cannot verify the ethical review approval record as described. If the authors can provide clarification on the ethical review approval, I am happy to recommend the acceptance of this paper.

**Strengths:**

1. **Valuable Data**: The PEACE dataset is a significant addition to the field due to its comprehensive and long-term pain records, which are scarce in existing datasets. While this data can be noisy, it provides valuable insights into pain management over time. Additionally, the geographical diversity of the dataset, which contrasts with the predominantly US-focused datasets, helps reduce potential biases in medical treatment and concepts.
2. **Effective De-identification**: The authors have taken necessary steps to de-identify patient data, particularly demographic data that could be directly traced back to patients. Although some medical history data could be considered protected health information, the descriptions are vague enough to maintain privacy.

**Additional Feedback:**

N/A

**Clarity:**

- The manuscript is generally clear.

**Correctness:**

- The content appears generally correct.

**Documentation:**

- The explanation of numerical and categorical features is clear and reasonable. The paper also provides detailed descriptions of de-identification and data characteristics. However, due to ethical constraints, access to the dataset requires an application, which I did not pursue to avoid violating peer-reviewing protocols.

**Ethics:**

- I could not find the ethical review approval record. I checked all available announcements of the Institutional Review Board of Xiangya Hospital through their [webpage](https://zndxyxrw.csu.edu.cn/yxll/zndxxyyyyxllwyh/llsc.htm). However, I did not find a record with approval ID 202109422 that matches the description in the dataset document. More information about the ethical review approval is needed for verification.

**Limitations:**

1. **Data Quality Assessment**: Due to ethical constraints, the dataset requires an application for access. To adhere to peer-reviewing protocols, I did not apply to access the dataset. Consequently, it is challenging to assess the data quality fully.

**Opportunities For Improvement:**

1. **Clarity on Medical Professional Rationales**: The paper mentions in line 53 that PEACE incorporates medical professionals' rationales behind medication plans, but this aspect is not clearly presented in the manuscript. Providing detailed examples or a clearer explanation would enhance understanding.

**Relation To Prior Work:**

- The description of related work is reasonable and comprehensive.

**Summary And Contributions:**

The paper presents the PEACE dataset, a comprehensive resource designed to enhance cancer pain management through pharmaceutical care. It includes records for over 38,000 patients and captures extensive data, including demographics, clinical examinations, treatment outcomes, medication plans, and patient self-perceptions. The dataset is particularly valuable as it incorporates multidisciplinary treatment (MDT) team assessments and long-term follow-up, both of which are often missing in existing datasets. The authors also demonstrate the potential of the dataset through proof-of-concept studies using 11 machine learning algorithms for treatment effectiveness assessment (TEA) and medication recommendation (MR).

---

> ### Author Rebuttal · Authors · 2024-08-16
>
> Thank you for your time, feedback, and insights. Please see the responses provided below.
>
> >Improvement:
>
> **Medication Plan Rationale:**
> Thank you for your valuable feedback and we would like to take the opportunity to further clarify the rationale behind the medication plans of healthcare professionals in the PEACE dataset.
>
> When a patient seeks treatment for pain relief, there are two possible scenarios: The first scenario involves a patient experiencing pain for the first time, with no prior use of analgesic drugs. The second scenario involves a patient already using analgesic drugs but seeking adjustments due to inadequate pain relief or intolerable side effects.
>
> For patients in the second scenario—those seeking adjustments—the PEACE dataset records their previous medication-related issues. This includes existing or potential drug-related problems (DRPs), the reasons behind these issues, the interventions taken, and the outcomes. These features are captured in the "Evaluation of Previous Analgesic Treatment" section of the dataset and are standardized according to the Pharmaceutical Care Network Europe (PCNE) Classification for DRPs. This classification reflects the healthcare professionals' reasoning for modifying pain medication. For example, if a patient on opioid medication experiences severe side effects such as constipation, the medical team may decide to alter the medication or introduce adjunctive treatments to manage these side effects. The rationale behind such adjustments is thoroughly documented in the dataset.
>
> However, for patients experiencing pain for the first time who have not used any analgesic drugs, there are no DRPs related to previous analgesic use, and therefore, the reasons behind their initial medication plans are not recorded in the same way. In these cases, the medication plan is developed by the medical team—usually a multidisciplinary treatment team (MDT)—based on the patient’s pain assessment results and established cancer pain treatment guidelines, such as those from the National Comprehensive Cancer Network (NCCN), the European Society for Medical Oncology (ESMO), and guidelines from China’s National Health Commission. For instance, if a patient’s pain intensity reaches a severe level, the team might consider strong opioids as a first-line treatment, following the three-step ladder approach for cancer pain. These decisions are guided by clinical experience and guideline recommendations, but the specific reasons behind the initial medication plans are not recorded in the dataset.
>
> >Limitations:
>
> **Data Quality:** Due to ethical considerations, access to this dataset requires an application process. We plan to initiate this process once our paper is accepted. Meanwhile, we have provided sample cases in the Supplementary Material for reference.
>
>
>
> >Ethics:
>
> **Ethical Approval:**
> Thank you for your inquiry regarding the ethical review approval for our project. Our project received ethical approval in 2021, but the Institutional Review Board (IRB) website of Xiangya Hospital currently displays the records only from July 2022 to December 2023. As a result, our approval, which was initially granted for two years starting in 2021 and later extended by one year until September 2024, does not appear within the current display range on their website.
>
> To provide full transparency, we have **attached the relevant ethical approval documents to the rebuttal**. Additionally, these documents have been uploaded to our project's GitHub homepage under the "Ethics Documents" section for the ethical reviewers to review.

---

> > ### Comment · Reviewer_HMu4 · 2024-08-21
> > **Response to authors' rebuttal**
> >
> > Thank you for the response. After reviewing the ethical approval documents, I find them satisfactory. I no longer have any concerns and have adjusted my score accordingly.

---

> > > ### Author Response · Authors · 2024-08-22
> > >
> > > We are glad to hear that your concerns were addressed. Thank you for the time and thoughtful review that helped us improve our submission. Please feel free to reach out if you have any further suggestions, feedback, or requests.

---

> ### Author Rebuttal · Authors · 2024-08-21
>
> Dear Reviewer HMu4,
>
> This is a friendly reminder that we have provided the requested ethical approval documentation in our rebuttal, along with detailed answers to your other questions. We would greatly appreciate your feedback on our responses and hope that they have addressed your concerns.
>
> We look forward to hearing your thoughts.

---

### Official Review · Reviewer_e7K2 · 2024-07-23
**Review of submission 715**

**Rating:** 6
**Confidence:** 4
**Correctness:** Yes
**Clarity:** Yes

**Review:**

The manuscript also details the construction of the dataset and includes a proof-of-concept study using 11 machine learning algorithms. These algorithms range from basic machine learning and neural network models to tree-based models and advanced neural network models designed for time-series data. The study evaluates treatment effectiveness and medication recommendations.

**Strengths:**

The sample size is relatively large for pain management data.

**Additional Feedback:**

Please refer above

**Documentation:**

In the feature description section, it would be beneficial to show each category for multi-class features.

**Ethics:**

Would there be a race information?

**Limitations:**

The manuscript mentions that advanced neural network models performed poorly, which is attributed to overfitting. However, there is no discussion on the steps taken to mitigate overfitting, such as regularization techniques or cross-validation strategies. Otherwise, the authors should try some advanced neural network models which requires fewer samples.

**Opportunities For Improvement:**

it may be beneficial to explore EHR-specific predictive models such as RETAIN, StageNet, Dr. Agent, AdaCare, ConCare, and GRASP.

**Relation To Prior Work:**

Yes

**Summary And Contributions:**

The manuscript presents the PEACE dataset, a comprehensive resource designed for pharmaceutical care in cancer pain management. This dataset includes detailed pharmacological care records for over 38,000 patients, covering demographics, clinical examinations, treatment outcomes, medication plans, and patient self-perceptions. Unlike existing datasets, PEACE incorporates long-term follow-ups both inside and outside hospitals, and includes multidisciplinary treatment (MDT) team assessments as well as patients’ self-assessments.

---

> ### Author Rebuttal · Authors · 2024-08-16
>
> Thank you for your comments.
>
> >Improvement:
>
> **Explore More Prediction Models:**
> Thanks for the extra benchmarks suggestion, we managed to evaluate these models, including RETAIN[1], StageNet[2], Dr. Agent[3], AdaCare[4], ConCare[5], and GRASP[6], performance on electronic health records (EHR). Our experimental results show that these models not only outperform general neural network models in modeling EHR data with time series, but also show higher stability in cross-validation. Some of the performance metrics of these models have exceeded existing SOTA levels. Additionally, some methods, like Dr. Agent and ConCare, were originally developed using an earlier version of the MIMIC dataset. Their data processing steps had not been updated to align with the latest MIMIC data. As a result, we adjusted these methods to ensure consistency with the format of our dataset.
>
> The experimental results are attached in the **attachment** of the rebuttal, the newly added text is highlighted in RED.
>
> **References:**
>
> [1] Choi, Edward, et al. "Retain: An interpretable predictive model for healthcare using reverse time attention mechanism." Advances in neural information processing systems 29 (2016).
>
> [2] Gao, Junyi, et al. "Stagenet: Stage-aware neural networks for health risk prediction." Proceedings of The Web Conference 2020. 2020.
>
> [3] Gao, Junyi, et al. "Dr. Agent: Clinical predictive model via mimicked second opinions." Journal of the American Medical Informatics Association 27.7 (2020): 1084-1091.
>
> [4] Ma, Liantao, et al. "Adacare: Explainable clinical health status representation learning via scale-adaptive feature extraction and recalibration." Proceedings of the AAAI Conference on Artificial Intelligence. Vol. 34. No. 01. 2020.
>
> [5] Ma, Liantao, et al. "Concare: Personalized clinical feature embedding via capturing the healthcare context." Proceedings of the AAAI Conference on Artificial Intelligence. Vol. 34. No. 01. 2020.
>
> [6] Zhang, Chaohe, et al. "GRASP: generic framework for health status representation learning based on incorporating knowledge from similar patients." Proceedings of the AAAI conference on artificial intelligence. Vol. 35. No. 1. 2021.
>
> >Limitations:
>
> **Overfitting Issue:**
> Thank you for your feedback regarding the potential overfitting issue. We would like to clarify how our approach addresses this concern. As described on line 257 of the manuscript, the dataset was randomly divided into a training set and an independent test set in an 80/20 ratio. We then performed 5-fold cross-validation on the training set to ensure robustness. The captions for Tables 4 and 5 state that the values represent the average results from the 5-fold cross-validation runs, along with their mean errors. Detailed results for each fold are provided in Tables 14 and 15 in the Appendix. Additionally, to further mitigate the risk of overfitting during model training, we implemented an Early Stopping mechanism, where training was halted if no improvement was observed within 10 epochs.
>
>
>
>
> >Ethics:
>
> **Race Information:**
> Due to privacy concerns, race information has not been included in the current version of this dataset. However, as we continue to develop and refine the dataset, we will carefully consider this request and explore the possibility of including such information in a future version.

---

> > ### Comment · Reviewer_e7K2 · 2024-08-20
> > **Response to author rebuttal**
> >
> > Thanks to the authors for their clarification and for including additional prediction models in the tasks. After reviewing the updates, I have decided to keep the score unchanged.
> >
> > I also hope that the authors will ensure the long-term maintenance of the dataset and guarantee its continued public accessibility.

---

> > > ### Author Response · Authors · 2024-08-21
> > > **Thank you for reading revisions & reply**
> > >
> > > Thank you for reading all of our updated content and replies, we are very pleased it was able to satisfy your concerns. Please do not hesitate to let us know if you had any further questions.

---

### Official Review · Reviewer_ofqK · 2024-08-08
**Review comments**

**Rating:** 7
**Confidence:** 4
**Correctness:** The claims are correct.
**Clarity:** The paper is well written.

**Review:**

For the NeurIPS audience, the primary contribution lies in the collection of the dataset. However, my main concern is the public accessibility of the dataset. Here are my detailed comments:

1. The authors have not established an online application and review platform similar to PhysioNet. They state that external researchers will be reviewed by the Xiangya Hospital’s Big Data Management Center, but it is unclear how stringent the review process is, especially for global researchers outside of China. Based on my experience, it can be quite challenging for medical institutions to release sensitive data to external, particularly international, users.
2. What is the temporal data distribution? The authors mention that the dataset "includes multi-visit, long-term observations for 2,600 patients." Does this imply that only these 2,600 patients have multiple visits while the rest have data from a single visit? The data distribution (e.g., average number of visits per patient) should be reported.

The main contribution of this paper is the introduction of a new dataset. Therefore, my primary concern revolves around how the accessibility of this dataset is ensured for researchers. Generally, I believe that journals focusing on medical and data science, such as Scientific Data, might be more suitable for this work, as they have more stringent rules to ensure public accessibility.

**Strengths:**

The dataset is comprehensive and clinically meaningful. I appreciate the effort of inviting clinical experts, which could be very helpful for medical research.

**Additional Feedback:**

NA

**Documentation:**

There are a few instructions. But the detailed documents are not provided.

**Limitations:**

Please see my comments above.

**Opportunities For Improvement:**

Please see my comments above.

**Relation To Prior Work:**

Prior works are discussed.

**Summary And Contributions:**

In this work, the authors have collected and released a new dataset for evaluating pharmaceutical care in cancer pain management. The dataset includes records of over 38,000 patients. Clinical experts were also invited to assess the health status of these patients. The dataset is comprehensive and clinically meaningful.

---

> ### Author Rebuttal · Authors · 2024-08-16
>
> We appreciate Reviewer ofqK's valuable comments. Please find our response below.
>
> >Comment 1:
>
> **Application Review:**
> Regarding your inquiry about our online application and review platform, we have not yet established a system similar to PhysioNet. However, our GitHub page provides clear guidelines for data usage, download instructions, and contact information, allowing users to easily complete the application process by following the provided steps. Our data application and review process is modeled after PhysioNet, with our user agreement based on best practices from multiple medical datasets, such as the MIMIC database. Once an application is approved, we grant access to the dataset.  Similarly, in the NeurIPS 2021 Datasets and Benchmarks Track, Ordun et al. [1] developed a dataset of chronic cancer pain facial tests collected from clinical trials at the NIH involving 29 patients. Data access was also provided through direct contact with the authors and obtaining NIH authorization.
> We plan to develop a more comprehensive online application platform in the future.
>
> To ensure data security and compliance, applicants must be affiliated with an academic or industrial institution and provide proof of relevant legal training, such as HIPAA (Health Insurance Portability and Accountability Act) certification. Data access is restricted to the applicant and cannot be redistributed. All applicants must obtain approval from Xiangya Hospital for relevant legal and research agreements. In the data usage agreement, users are required to provide accurate personal information and clearly state their research purpose. Internal researchers affiliated with Xiangya Hospital or those with connections to it will receive expedited approval for quicker access to the data. These access restrictions are necessary to comply with Chinese law, Institutional Review Board (IRB) approval, and the high sensitivity of cancer patient data.
>
> **International User Accessibility:**
> In response to concerns about international access to Chinese medical data, we would like to highlight that several existing studies have successfully utilized data from Chinese medical institutions. For example, Zeng, Xian, et al.[2] introduced a large pediatric dataset from the Children’s Hospital of Zhejiang University, School of Medicine in China. This dataset was extracted from various hospital information systems, including the electronic medical records system, laboratory information system, computerized physician order entry system, nursing information system, anesthesia information management system, and reporting systems from departments like radiology, ultrasound, ECG, pathology, and more. The researchers emphasized that this project did not interfere with clinical care and that all protected health information was anonymized, eliminating the need for individual patient consent. After thorough de-identification, the dataset was made publicly available.
>
> Zhang, Zhongheng, et al.[3] presented a heart failure patient dataset from the Fourth People's Hospital of Zigong in Sichuan Province, China. This dataset integrates electronic medical records and follow-up outcomes, comprising information on 2,008 heart failure patients across 166 features.
>
> Xu, Ping, et al.[4] introduced an ICU dataset from the Fourth People's Hospital of Zigong, China, covering patients aged over 18 who were admitted to the ICU between January 2019 and December 2020. After adequate de-identification, this dataset was also made publicly available.
>
> **References:**
>
> [1] Ordun, Catherine. "Intelligent sight and sound: A chronic cancer facial pain dataset." Thirty-fifth Conference on Neural Information Processing Systems Datasets and Benchmarks Track. 2021.
>
> [2] Zeng, Xian, et al. "PIC, a paediatric-specific intensive care database." Scientific data 7.1 (2020): 14.
>
> [3] Zhang, Zhongheng, et al. "Electronic healthcare records and external outcome data for hospitalized patients with heart failure." Scientific data 8.1 (2021): 46.
>
> [4] Xu, Ping, et al. "Critical care database comprising patients with infection." Frontiers in Public Health 10 (2022): 852410.
>
>
> >Comment 2:
>
> **Data Distribution:**
> The PEACE dataset comprises records from 38,766 patients, of whom 2,601 (approximately 7\%) have multiple visits, while the remainder have data from only a single visit.
>
> In this dataset, each new patient visit is recorded, allowing us to observe long-term patient interactions with Xiangya Hospital. Consequently, some patients have accumulated multiple visits over several years, forming extensive longitudinal records. The minimum number of visits recorded for any patient is 1, with a maximum of 33 visits. On average, a patient has 1.09 visits. For patients with multiple visits (i.e., 2 or more), the average rises to 2.48 visits, with a range from 2 to 33 visits. As we increase the threshold for the number of visits, the average number of visits naturally increases, while the number of patients meeting that threshold decreases accordingly.
>
> The dataset contains a total of 42,638 records collected between 2016 and 2023. The annual distribution of these records is as follows: 60 records in 2016, 521 in 2017, 807 in 2018, 815 in 2019, 11,828 in 2020, 4,431 in 2021, 13,719 in 2022, and 10,457 in 2023.

---

> > ### Comment · Reviewer_ofqK · 2024-08-16
> > **Reviewer's reply**
> >
> > Thanks to the authors for their replies. To clarify, I don't have specific concerns about the accessibility of Chinese medical data, but most healthcare institutions globally face data accessibility issues for external users. Therefore, it would be helpful if the authors could provide details about the internal review pipeline (e.g., What forms or documents the applicants need to provide, how long the general review time is estimated, etc.). I believe such information could be very helpful for relevant researchers to use this dataset.

---

> > > ### Author Rebuttal · Authors · 2024-08-18
> > >
> > > Thank you for the reviewer’s prompt response and for raising additional questions to help us further improve the quality of our paper. We appreciate the reviewer’s clarification and apologize for any confusion in our previous response. Some of the information regarding the internal review pipeline had provided in the appendix, so our focus was primarily directed towards International User Accessibility.
> > >
> > > As indicated in Appendix Section E, lines 743-747, we briefly described the data access process, with more detailed information available at our GitHub repository - https://github.com/YTYTYD/PEACE. As outlined on our GitHub homepage, the access process involves three steps:
> > >
> > > 1) Complete some training such as the CITI (Collaborative Institutional Training Initiative at the University of Miami) “Data or Specimens Only Research” course as an MIT affiliate, as described in the instructions for completing required CITI training. Or you could provide a GCP certification;
> > >
> > > 2) Carefully read the terms of the **Data Use Agreement** (Please note that this agreement is a Word document available for download on our GitHub website. It contains all the information required for the data access review.), if you agree and wish to proceed, please send your application to the manager (Jian Xiao). Please use an official e-mail address such as .edu;
> > >
> > > 3) Final approval of data access is required by Xiangya Hospital's Big Data Management Center.
> > >
> > > Once an application has been approved, the researcher will receive emails containing instructions for downloading the dataset.
> > >
> > > Currently, we estimate a response time of 20 business days for processing requests. This duration may vary depending on the completeness of the provided information and can take up to three months. However, we are actively working on automating the approval process and aim to reduce the turnaround time to 7 business days.
> > >
> > > We will ensure that all the above information is clearly highlighted in the main text of the revised paper.

---

> > > > ### Comment · Reviewer_ofqK · 2024-08-19
> > > > **Response to author rebuttal**
> > > >
> > > > Thanks to the authors for their clarification. I've updated my score. I believe a public medical dataset with expert labeling is valuable for this field. I hope the authors can ensure long-term maintenance of the dataset.

---

> > > > > ### Author Response · Authors · 2024-08-21
> > > > > **Thank you for update**
> > > > >
> > > > > Thank you for your positive feedback and for recognizing the value of the PEACE dataset. We greatly appreciate your support and will continue to prioritize the development and maintaince of the PEACE dataset for advancing research in this field. Thank you once again for your thoughtful review and encouragement.

---

### Official Review · Reviewer_Azjb · 2024-08-08
**Review for Paper #715**

**Rating:** 7
**Confidence:** 4
**Correctness:** Yes
**Clarity:** Yes

**Review:**

The PEACE dataset from China seems valuable, but it is still not publicly released and requires an application instead. The paper has limited technical contribution but with the relatively high data-side quality as described in the paper (sufficient patient volume and tasks and experts' feedback), I suggest submitting the paper to a more dataset-centric and medically relevant journal like *Scientific Data*. Additionally, the paper claims PEACE incorporates medical professionals' assessments of the current health state and the rationale behind medication plans, which are not present in existing datasets. However, I did not find evaluations or case studies for this section in the main paper (the utilized MDT seems only for feature selection, rather than the claimed assessments of the current health state and the rationale behind medication plans), while it only benchmarked on defined tasks classification or regression. Apart from this, I generally agree with Review #ofqK's questions and reviews.

**Strengths:**

see Review

**Additional Feedback:**

N.A.

**Documentation:**

Yes

**Limitations:**

see Review

**Opportunities For Improvement:**

see Review

**Relation To Prior Work:**

Yes

**Summary And Contributions:**

The paper introduces PEACE (Pharmaceuticals for Easing Cancer Pain with Care), a comprehensive dataset specifically designed for cancer pain medication research. PEACE contains over 38,000 patient records, with 103 features related to various pathologies, symptoms, and etiologies; provides long-term, multi-visit observations for 2,600 patients, offering further insights into patient care trajectories; incorporates medical professionals' assessments and rationale behind medication plans; and includes extensive experiments validating 11 machine learning and deep learning approaches for treatment effect evaluations and medication decision-making.

---

> ### Author Rebuttal · Authors · 2024-08-16
>
> Thank you for your thoughtful comments. We noticed that this review appeared late in the system, but we have made every effort to address your concerns thoroughly. If any part of our responses remains unclear, we would be glad to provide further clarification during the reviewer-author communication period.
>
> >Venue Suggestion:
>
> Our research is motivated by the desire to explore the application of machine learning techniques in real-world scenarios, particularly within the medical field. Much of the existing research relies on synthetic data or data generated in controlled laboratory environments, which often fail to capture the full complexity and unpredictability of real-world situations. Collecting medical data, in particular, involves navigating complex ethical issues, adding further layers of challenge. Our dataset is sourced from an actual clinical setting and effectively demonstrates the potential of machine learning when applied to real, complex, and variable medical data. We believe our work is well-suited for presentation at conferences like NeurIPS, as it provides valuable insights for ML researchers focused on AI for social good.
> Similarly, in the NeurIPS 2021 Datasets and Benchmarks Track, Ordun et al. [1] developed a dataset of chronic cancer pain facial tests collected from clinical trials at the NIH involving 29 patients. Data access was also provided through direct contact with the authors and obtaining NIH authorization.
>
> >Effects of MDT:
>
> Before addressing the issue regarding the MDT's assessment of patients' health status and the rationale behind the drug plans in the dataset, we would like to clarify the feature selection process. We involved 32 experts from various regions across the country and employed the Delphi technique. This method allows experts to anonymously express their opinions and gradually reach a consensus through multiple rounds of feedback. It’s important to note that these experts were independent and not part of an actual MDT team.
>
> In our dataset, all evaluations and decisions were made by MDT teams, typically comprising a clinician and a pharmacist, and occasionally including a nurse, during actual clinical practice. We recorded the data generated while providing medical care to cancer pain patients. The 32 medical experts from different disciplines were then invited to select the features most significant for cancer pain medication decision-making, thereby constructing the dataset.
>
> Here’s an overview of the data collection process:
>
> When a patient seeks pain relief at the hospital, the doctor first gathers the patient's basic information (e.g., name, age, diagnosis, physical status score). The pharmacist then assesses the patient's pain and reviews the rationality of any previous medication plan, identifying potential drug-related problems (DRPs) and analyzing their causes. The MDT, including both the doctor and pharmacist, discusses these findings and adjusts the medication plan accordingly. The patient's information is entered into a follow-up system, which tracks the effectiveness of the intervention and allows ongoing communication with healthcare providers. This comprehensive data, including initial evaluations, medication adjustments, and follow-up results, informs our feature selection.
>
> For example, in Sample 4 of the manuscript (Table 19), the data collection begins with the clinician recording the patient’s basic information. The pharmacist then evaluates the patient’s pain and the effectiveness of the current medication, noting that the patient, suffering from severe somatic pain, was inadequately managed with NSAIDs. According to the WHO cancer pain ladder principle [2], the MDT adjusted the medication plan to include strong opioids combined with NSAIDs [3, 4]. After one week, the patient's pain was fully relieved, demonstrating the effectiveness of the MDT's decision-making process.
>
> The MDT’s rationale behind medication adjustments is captured in the "Evaluation of Previous Analgesic Treatment" feature set. This includes standardized features such as existing/potential DRPs, causes, interventions, execution of the intervention plan, and final status of DRPs, following the Pharmaceutical Care Network Europe (PCNE) Classification for DRPs. These features document the decision-making process from identifying drug-related problems to implementing and assessing treatment outcomes.
>
> **References:**
>
> [1] Ordun, Catherine. "Intelligent sight and sound: A chronic cancer facial pain dataset." Thirty-fifth Conference on Neural Information Processing Systems Datasets and Benchmarks Track. 2021.
>
> [2] World Health Organization: Cancer Pain Relief: with a Guide to Opioid Availability; 1996.
>
> [3] Swarm RA, et al.: Adult Cancer Pain, Version 3.2019, NCCN Clinical Practice Guidelines in Oncology. J Natl Compr Canc Netw 2019, 17(8):977-1007.
>
> [4] National Health Commission of the People’s Republic of China: Standard Diagnosis and Treatment of Cancer Pain (2018). Chinese Clinical Oncology 2018, 23(10):937–944.

---

> > ### Author Rebuttal · Authors · 2024-08-17
> >
> > >Limited Technical Contribution:
> >
> > As stated in the NeurIPS 2024 Datasets and Benchmarks Track [5], this track does not require "algorithmic advances." It is recommended [6] to mention "algorithmic advances" in submissions to the main conference. We adhered to the scope of this track [7], which emphasizes "new datasets and benchmarks on new or existing datasets." Thus, our dataset and benchmark paper does not propose new deep learning methods.
> >
> > The main contributions of our paper are as below:
> > First, we identified the limitations of existing datasets, which fail to provide long-term patient observations and fully encompass all the information required for MDT decision-making. To address these limitations, we proposed a new comprehensive cancer pain medication dataset, PEACE, which includes over 38,000 patients experiencing cancer-related pain, among whom more than 2,600 have multiple long-term follow-up records. Using this dataset, we evaluated the performance of 11 models(Increased to 17 models during rebuttal) in classification and regression tasks, aiming to advance research in machine learning.
> >
> > Moreover, compared to the purely clinical databases in Scientific Data, the innovation of the PEACE dataset lies not only in the breadth and depth of its data but also in the new challenges it poses to the field of machine learning, particularly in the development of personalized treatment effect evaluation and medication recommendation systems. In conclusion, we believe that the PEACE dataset aligns with the objectives of the NeurIPS 2024 Datasets and Benchmarks Track, promoting innovation in machine learning research and driving algorithmic development.
> >
> > **References:**
> >
> >
> > [5] NeurIPS 2024/ Call For DatasetsBenchmarks
> >
> > [6] If the main contribution is a new dataset, benchmark, or other work that falls into the scope of the track (see above), then it is ideally reviewed accordingly. As discussed in our blog post, the reviewing procedures of the main conference are focused on algorithmic advances, analysis, and applications, while the reviewing in this track is equally stringent but designed to properly assess datasets and benchmarks. Other, more practical considerations are that this track allows single-blind reviewing (since anonymization is often impossible for hosted datasets) and intended audience, i.e., make your work more visible for people looking for datasets and benchmarks.
> >
> > [7] "SCOPE". In addition to new datasets and benchmarks on new or existing datasets. we welcomesubmissions that detail advanced practices in data collection and curation that are of general interest even ithe data itself cannot be shared, Data generators, reinforcement learning environments, or benchmarkingtools are also in scope, Frameworks for responsible dataset development, audits of existing datasets,identifying significant problems with existing datasets and their use, or systematic analyses of existingsystems on novel datasets that yield important new insight are also in scope.

---

> > > ### Comment · Reviewer_Azjb · 2024-08-19
> > >
> > > **Thanks for your response!** I appreciate that you've integrated the rigorous MDT process, but I’m still not entirely clear on how the PEACE dataset reflects these insights to facilitate medication recommendations, simulate professionals' rationale, evaluate personalized treatment effects, and so on. Could you clarify whether there is any structured or unstructured information within the dataset that supports these points? For example, if someone were to conduct research related to medication recommendations, would they be able to use the PEACE dataset for this purpose?
> > >
> > > In other words, if the PEACE dataset is more than just a longitudinal EHR dataset where the MDT process is used for feature selection, what additional information does it provide?
> > >
> > > > particularly in the development of personalized treatment effect evaluation and medication recommendation systems
> > >
> > > > PEACE incorporates medical professionals’ assessments of the current health state and the rationale behind medication plans, which are not present in existing datasets
> > > ---
> > > I've increased the score, but I hope to receive a more detailed answer. If my concerns are addressed, I will continue to raise the score further.

---

> > > > ### Author Rebuttal · Authors · 2024-08-21
> > > >
> > > > Thank you for your thoughtful feedback and for giving us the opportunity to further clarify and improve our paper.
> > > >
> > > > >The Unique Features of the PEACE
> > > >
> > > > The PEACE dataset is designed specifically for pharmaceutical care in cancer pain management, going beyond the capabilities of traditional longitudinal EHR datasets. Unlike existing datasets, such as the cancer pain dataset [1], which focuses on pain assessment through facial expressions, the PEACE dataset incorporates a broader range of structured data, including physiological features, self-assessments, and expert evaluations determined through the Delphi method by a MDT team. This structured data is directly relevant to clinical decision-making and has been carefully curated to reflect the complexities of cancer pain management.
> > > >
> > > > The PEACE dataset includes:
> > > > 1) **Structured Data:** All features are organized into binary, multi-class, and numerical types, making them directly applicable to machine learning algorithms. This data includes detailed patient information, pain assessments, previous analgesic treatments, treatment evaluations, and follow-up outcomes. These structured features can be used to simulate professional decision-making and support the development of medication recommendation systems.
> > > >
> > > > 2) **Unstructured Data:** While the current version of the PEACE dataset primarily contains structured data, we plan to incorporate unstructured data, such as free-text feedback from patients and clinicians' notes, in future updates. This unstructured data will provide deeper insights into the rationale behind medication decisions and offer additional context for personalized treatment evaluations.
> > > >
> > > > **References:**
> > > >
> > > >
> > > > [1] Ordun, Catherine. "Intelligent sight and sound: A chronic cancer facial pain dataset." Thirty-fifth Conference on Neural Information Processing Systems Datasets and Benchmarks Track. 2021.
> > > >
> > > >
> > > > >Relevance to Treatment Effectiveness Assessment and Medication Recommendations
> > > >
> > > > The PEACE dataset is designed to align closely with practical medical needs, particularly in ongoing cancer pain management, where patients continue medication long after discharge. The inclusion of extramural follow-up data allows for tracking patients' treatment experiences outside the hospital, including medication continuation at home and any adverse reactions. This is crucial for personalized medication adjustments.
> > > >
> > > > For example, pain is a subjective experience, and standardized scales may not capture the full impact on a patient’s life. The PEACE dataset records how pain affects seven aspects of a patient's life (daily living, emotions, mobility, work, relationships, sleep, and interest in life). This information is critical for evaluating pain relief effectiveness and adjusting treatment accordingly. Features like breakthrough pain occurrence influence immediate-release medication choices, which are recorded in the Comprehensive Pain Assessment section in lines 570-583 of the paper.
> > > >
> > > > From the doctor's perspective, features such as patient compliance are assessed using the MMAS-8 scoring system (Previous Analgesic Treatment section, lines 608-625). Higher scores indicate better adherence, leading to potentially better pain relief outcomes.
> > > >
> > > > >Decision-Making and Dataset Utility
> > > >
> > > > Our retrospective data analysis of treatment plans and outcomes enables a comprehensive understanding of different methods' effectiveness, promoting innovation in machine learning algorithms. These features allow researchers to model professional rationale and develop more accurate medication recommendation systems.
> > > >
> > > > Moving forward, we plan to expand the PEACE dataset, similar to the MIMIC dataset, by incorporating unstructured data, such as free-text notes and genetic information (e.g., CYP2D6, CYP3A5, OPRM1, and TAOK3). These additions will further support personalized treatment and medication recommendation research, although genetic data will require extended ethical approval.
> > > >
> > > > We hope this response clarifies how the PEACE dataset is more than just a longitudinal EHR dataset and how it can serve as a valuable resource for researchers and developers working on personalized treatment and medication recommendation systems.

---

> ### Comment · Reviewer_Azjb · 2024-08-21
>
> Thanks for you further clarification. You have fully addressed my concerns, I've raised the score.

---

> > ### Author Response · Authors · 2024-08-22
> >
> > We would like to express our gratitude once again for your insightful feedback, which was instrumental in helping us revise our manuscript for better clarity.

---

### Author Rebuttal · Authors · 2024-08-16

We sincerely thank each reviewer for their valuable time and feedback. Before reviewing our responses, we kindly ask you to download the attached PDF file.

Regards,

Authors

---

### Decision · Program_Chairs · 2024-09-26

**Decision:**

Accept (Poster)

**Comment:**

This paper introduces a novel dataset (PEACE) designed specifically for cancer pain medication research. PEACE comprises over 38,000 patient records with 103 features related to various pathologies, symptoms, and etiologies. It offers long-term, multi-visit observations for 2,600 patients, providing deeper insights into patient care trajectories. The dataset also includes medical professionals' assessments and rationales behind medication plans, alongside extensive experiments validating 11 machine learning and deep learning approaches for evaluating treatment effects and supporting medication decision-making.

While the dataset is comprehensive and clinically valuable, the authors should clarify how the dataset can be accessed (perhaps in Section 3.7) and include additional baseline results as mentioned in the rebuttal. Furthermore, the reviewers encourage the authors to continuously update the PEACE dataset, which will be beneficial for the broader research community working on machine learning for healthcare.